# Volatile resorption expedites eruption onset in large silicic systems

**Franziska Keller** [1,2] ✉**, Meredith Townsend** [1]**, Juliana Troch** [3] **& Christian Huber** [4]

Silicic caldera-forming eruptions are among the most hazardous natural phenomena on Earth, yet their triggering mechanisms remain poorly understood. While volatile exsolution is widely recognized as a potential eruption driver in large silicic systems, we find that volatile resorption can, counterintuitively, promote chamber pressurization faster than volatile exsolution. Using a thermo-mechanical magma chamber model, we show that resorption is a common process in rapidly recharged systems, driven by pressure increase and crystal melting. The Aso-4 eruption offers a natural case where volatile resorption may have occurred, with model results predicting resorption at recharge rates $>10^{-2.4}$ km$^3$/yr. Through reducing bulk magma compressibility, resorption amplifies pressurization, driving chamber destabilization and potentially expediting eruption onset. Here, we propose that volatile resorption is a natural process both accommodating and promoting rapid chamber pressurization, fundamental to destabilizing large-scale silicic systems. Detecting its signatures in monitoring signals could provide early warning of imminent eruption.

Triggering silicic caldera-forming eruptions (≥VEI 7) requires extraordinary conditions, as pressurization of large-volume reservoirs is inherently difficult. Such extensive systems usually form through prolonged magma injection into the shallow crust over timescales exceeding $10^4$ years, leading to thermal maturation of the surrounding crust[1]. The associated reduction in crustal viscosity slows down internal pressure build-up, allowing magma chambers to grow to extreme sizes (sometimes ≥500 km$^3$) without erupting. As reservoirs grow to larger volumes, recharge-induced pressurization becomes less effective, since achieving critical overpressure for eruption requires proportionally higher recharge rates[2,3]. Consequently, eruption triggering in large-scale silicic reservoirs is significantly more challenging than in average-sized silicic systems[4,5].

Various triggering mechanisms have been proposed for eruptions at caldera-forming systems, which are broadly subdivided into two groups: (1) external processes, such as roof collapse or ice unloading, and (2) internal processes, like reservoir pressurization

through magma recharge or continued volatile exsolution[2,6–8]. Especially in water-rich silicic systems, volatile exsolution is often considered a main driver for internal pressurization inducing eruption[9–15]. In the early stages of accumulation, subvolcanic reservoirs can evolve without an exsolved magmatic volatile phase (MVP). Over time, magma differentiation and crystallization lead upper-crustal reservoirs to volatile saturation and MVP exsolution[8,16–18]. This process increases overpressure as the low-density MVP causes magma to expand in volume against stiffer crustal rocks[7,11–15,19]. However, for volatile exsolution to act as primary eruption trigger, exsolution must outpace both volatile loss by passive degassing and viscous relaxation of the crust. This, however, requires rapid crystallization rates that are difficult to maintain in larger, thermally buffered reservoirs[2]. In large silicic systems, exsolved volatiles may therefore exert a primary control on magma compressibility and chamber growth, rather than directly triggering eruptions. The high compressibility of the MVP increases the bulk magma

[1]Department of Earth and Environmental Sciences, Lehigh University, Bethlehem, PA, USA. [2]Discipline of Geology, School of Natural Sciences, Trinity College Dublin, Dublin, Ireland. [3]Division of Geoscience and Geography, Faculty of Georesources and Materials Engineering, RWTH Aachen University, Aachen, Germany. [4]Department of Earth, Environmental, and Planetary Sciences, Brown University, Providence, RI, USA. ✉e-mail: kellerf@tcd.ie

compressibility, dampening pressurization from recharge and leading to less frequent, yet larger, eruptions[2,3,7,8,12,20,21].

We hypothesize that pre-eruptive MVP resorption, the opposite of volatile exsolution, is a naturally occurring process in large silicic systems that reduces magma compressibility, thereby modulating a system's response to recharge, its overall stability, and the timing of eruptions. Our hypothesis is motivated by a case of observed pre-eruptive MVP reduction at the Aso caldera in Japan[22]. In this system, water saturation levels prior to the Aso-4 caldera-forming event (>500 km$^3$ DRE, 86.4 ± 1.1 ka, 5–25 vol% crystallinity[23–25]) were reconstructed through a combination of melt inclusion data and volatile partitioning between apatite, melt, and a MVP. The study reveals a shift from water-saturated conditions during the Aso-Y event (-91.4 ka, 0.05 km$^3$), the last known eruption prior to Aso-4, to water-undersaturated conditions during Aso-4. Passive degassing of the MVP could be ruled out as a cause of the observed volatile reduction, as this processes is generally not efficient enough to remove all exsolved MVP[26,27]. Apatite would therefore continue to record water-saturated partitioning behavior rather than the observed under-saturated trends. This suggests that other processes must have contributed to reduction of the MVP. Instead, the volatile loss is linked to an apparent recharge event that replenished the system with a large quantity of more mafic magma within the last 5 kyrs before eruption. Here, we perform thermo-mechanical modeling informed with geo-chemical data from Keller et al.[22], to test whether volatile resorption could explain the observed MVP reduction and be a potentially over-looked, naturally occurring process. Using Aso as a case study, we aim to identify the physical and chemical conditions that promote volatile resorption and investigate whether it has unrecognized impacts on reservoir pressurization and eruption onset.

## Results and discussion

### Volatile dynamics prior to the Aso-4 caldera eruption

We use a 1D thermo-mechanical box model[2,3,21] to reconstruct volatile saturation levels in the 5 kyrs between the Aso-Y eruption and the Aso-4 caldera-forming event. This model simulates the thermal and mechanical evolution of magma chambers (the mobile part of a magma reservoir containing less than 50 vol% crystals) situated in a visco-elastic crust, where the residing magma is experiencing phase transitions such as crystallization and volatile exsolution. Chambers undergo continuous recharge and are subject to mass withdrawal via eruption once a critical overpressure (here 20 MPa) is reached. We calibrate the initial model setup to match pre-Aso-4 conditions by parametrizing a melting curve for rhyodacitic Aso-Y compositions (see methods section). The initial chamber volume is set to 500 km$^3$ [24] at 6 km depth[28,29] and the maximum model runtime is limited to 5 kyrs, reflecting the time interval between the Aso-Y and Aso-4 eruptions. The inflowing magma is hotter and contains lower $H_2O$ and higher $CO_2$ contents than the resident magma, to simulate pre-eruptive hybridization of the Aso-4 reservoir with less evolved recharge. To ultimately test if and under which conditions volatile resorption occurred, we vary initial conditions of the resident magma, such as crystallinity, $H_2O$ and $CO_2$ contents, as well as the temperature, injection rate, $H_2O$ and $CO_2$ contents of the recharging magma (Table 1).

As volatile budgets in apatite and melt inclusions show that the Aso-Y reservoir was water-saturated before its eruption[22], the presence of an MVP is required at the beginning of each simulation. We find that volatile saturation at the initial time step of our runs is achieved when a resident magma with 15 vol% crystals contains at least 4 wt% initial $H_2O$ (Fig. 1B). Runs with $H_2O$ below 4 wt% do not exsolve an initial MVP and are therefore excluded from further analysis. In the remaining cases, changes in volatile saturation are systematically linked to recharge-induced variations in crystal and MVP proportions that enable volatile exsolution as well as resorption. Based on these observations, we define four distinct regimes that describe the coupled thermal and

## Table 1 | Model parameters and initial conditions

| Variable | Parameter space | Aso-4 setup |
|---|---|---|
| Magma chamber volume [km$^3$] | 10, 100, 500, 1000 | 500 |
| Chamber depth [km] | 6, 8 | 6 |
| Initial magma crystallinity [vol%] | 2, 15 | 2, 15 |
| Initial $H_2O$ of resident magma [wt%] | 3, 3.5, 4, 4.5, 5 | 3, 3.5, 4, 4.5, 5 |
| Initial $CO_2$ of resident magma [ppm] | 100, 500 | 100, 500 |
| Recharge rate [km$^3$/yr] | $10^{-2}$ – $10^{-4}$ | |
| Recharge temperature [°C] | 900, 1300 | |
| Recharge $H_2O$ content [wt%] | 1, 4 | |
| Recharge $CO_2$ content [ppm] | 100, 500, 1000, 10,000 | |

Parameter ranges tested in this study for the broad range search and the Aso-4 model setup.

volatile saturation conditions in the chamber as a function of recharge rate: Regime 1 – Volatile Exsolution and Cooling, Regime 2 – Volatile Resorption and Cooling, Regime 3 – Volatile Resorption and Heating, and Regime 4 – Volatile Exsolution and Heating (Fig. 1A). For Aso, recharge rates ≤$10^{-3.1}$ km$^3$/yr allow volatile exsolution (second boiling) during reservoir cooling and crystallization (Regime 1, Fig. 1C). Rates >$10^{-3.1}$ km$^3$/yr initiate a transition to Regime 2, where the MVP is being resorbed, while reservoir cooling continues. Once supply rates increase, in our case exceeding ~$10^{-2.7}$ km$^3$/yr, recharge-induced heating drives crystal melting and simultaneous volatile resorption (Regime 3). Regime 4 is defined by volatile exsolution accompanied with recharge-induced reservoir heating, which is not observed in any of our runs. In the Aso simulations, eruption onset is achieved once recharge rates increase to at least $10^{-2.4}$ km$^3$/yr (Fig. 1C). We find that while recharge conditions such as higher temperatures (Fig. 1D, Supplementary Fig. 1B), lower $H_2O$ (Fig. 1E, Supplementary Fig. 1C) and lower $CO_2$ (Fig. 1F, Supplementary Fig. 1D) contents can promote resorption, recharge rate is the main control on the occurrence of this process (Fig. 1C, Supplementary Fig. 1A).

To achieve the transition from volatile saturation to under-saturation in a magma chamber as observed prior to Aso-4, the system must be at a sweet spot near saturation when the mass fraction of the initial MVP is small before the recharge event. For Aso, recharge rates >~$10^{-2.6}$ km$^3$/yr are required to achieve complete resorption of the MVP, which takes place at different $H_2O$ contents depending on the initial crystallinity: at 15 vol% magma crystallinity undersaturation is only reached in runs at 4 wt% initial $H_2O$ (Fig. 1C), while at 2 vol% initial crystallinity runs with up to 4.5 wt% initial $H_2O$ can undersaturate. These values match water contents (4.4–4.7 wt%) retrieved from plagioclase-melt hygrometry in Aso-4 magmas[25]. This overlap suggests that pre-Aso-4 conditions were ideally suited for recharge-driven volatile resorption, which potentially critically influenced the events leading up to the cataclysmic Aso-4 eruption.

### What drives volatile resorption during magma recharge?

As we find that recharge is the main driver for volatile resorption, we assess three processes linked to magma recharge as potential drivers for volatile resorption: (1) recharge-induced pressurization of the magma chamber, (2) mixing of silicic resident magma with less volatile-rich recharge magma, and (3) variations in phase proportions through crystal melting. Although these processes are not mutually exclusive, we aim to identify the hierarchy of processes driving volatile resorption by quantifying their individual contributions to variations in the MVP volume upon recharge. Therefore, we solve the equation for water mass conservation in a closed system and calculate the associated changes in the MVP volume fraction based on hypothetical scenarios (see Supplementary Methods). To evaluate the isolated

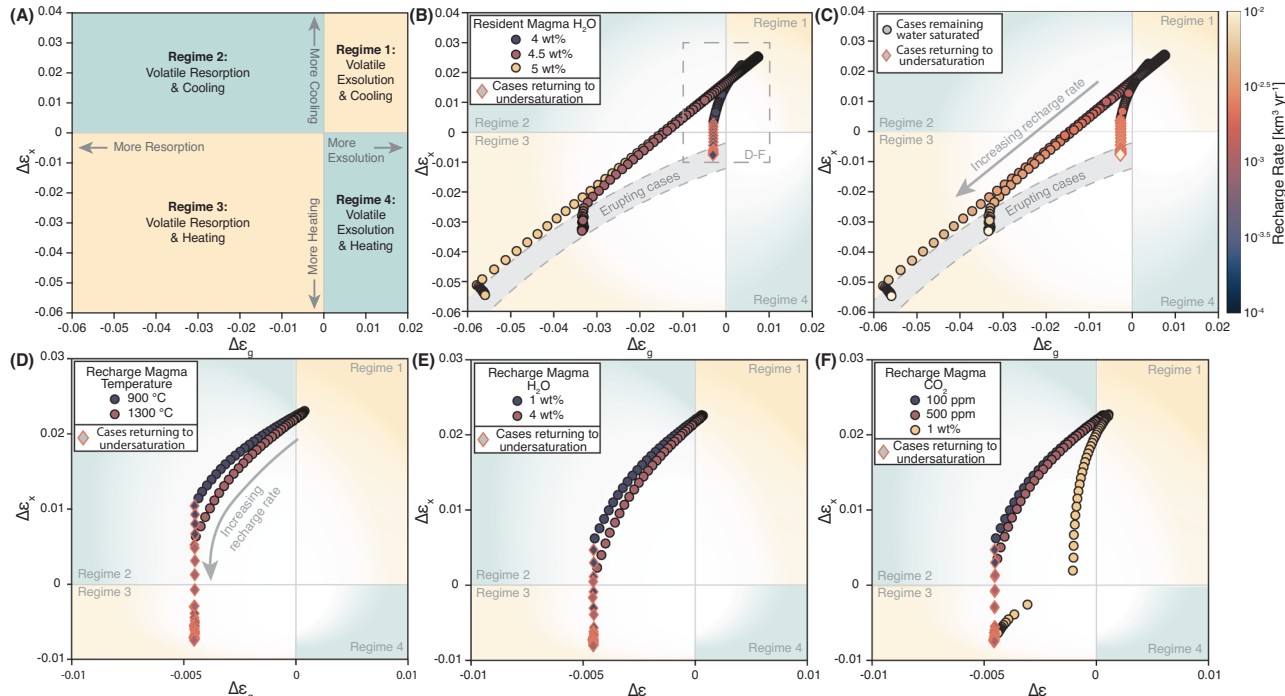

**Fig. 1 | Volatile and crystal behavior in Aso magma chamber simulations.**
**A** Regime diagram showing fields of volatile and crystal behaviors as a function of total changes in magmatic volatile phase (MVP) fractions ($\triangle\varepsilon_g$) and crystal fractions ($\triangle\varepsilon_x$) at the end of the simulations. Initial magma chamber conditions for runs depicted in panels (**B**)−(**F**) are set to 6 km depth, an initial volume of 500 km$^3$, 15 vol % initial magma crystallinity and 100 ppm $CO_2$. **B**−**C** show an identical subset of Aso data with variable initial $H_2O$ content in the resident magma (color-coded in **B**) and variable recharge rates (color-coded in **C**). Recharge conditions in these runs are kept constant at 1300 °C, 1 wt% $H_2O$ and 500 ppm $CO_2$. **D**−**F** show a close-up of the

dashed square in (**B**), representing cases with an initial resident magma $H_2O$ content of 4 wt%. Recharge conditions are varied to illustrate their impact on volatile and crystal behavior: **D** recharge temperature, **E** recharge $H_2O$, and **F** recharge $CO_2$. All plotted cases contain an initial MVP at the start of the run. The gray band labeled erupting cases highlights runs that underwent eruption before the end of the simulation. Diamond-shaped data points with pink rims indicate cases that transition from volatile-saturated to volatile-undersaturated conditions during the simulation.

effect of magma chamber pressurization resulting from high recharge rates (process 1), overpressure within the system is systematically varied between 0 and 25 MPa, while holding all remaining parameters fixed. As water solubility is a function of pressure, increasing overpressure values increases the solubility and therefore initiates resorption[12,30–33]. We find that such pressurization leads to significant resorption of the MVP of up to 4.3 vol%, corresponding to a relative volume decrease by 87 % assuming an initial MVP volume fraction of 5 vol% (Fig. 2A). However, as our simulations assume a fixed critical overpressure for eruption (20 MPa), the resulting pressurization effect, and thus the degree of volatile resorption, is constrained, defining a maximum extent to which resorption can occur. Under such constant critical overpressure thresholds, variations in the melt volume fraction (process 3) are suggested to account for significant variations in the MVP fraction. Here, influx of recharge introduces hotter melts into the previously evolved system, increasing the overall melt volume fraction, while also heating the chamber inducing crystal melting. The addition of anhydrous crystal melts and recharge melts dilutes the host magma's volatile content and drives it away from saturation. Equilibration of the now undersaturated melt with the still present MVP occurs through volatile diffusion from the MVP into the undersaturated melt, thereby reducing the MVP volume. Using the same calculations as above, we isolate the effect of changes in the crystal fraction by assuming no change in overpressure throughout the calculations. Instead, the values for crystal fraction are systematically varied from 10 vol% to 5 vol%. These changes in the melt volume fraction cause a loss of up to 2.4 vol% of the MVP, corresponding to a relative volume reduction by 47% (initial MVP volume fraction = 5 vol%, Fig. 2C). The effect of mixing between the resident and recharge

magmas (process 2), on the contrary, has a significantly smaller impact on changes in the MVP volume fraction. To simulate mixing of volatile-rich resident magmas with volatile-poorer (yet volatile-bearing) recharge magmas, we calculate the effects of increasing water mass and magma chamber mass for a hypothetical recharge scenario. The water mass to magma chamber volume ratio is calculated for a chamber volume increase of up to 1 vol%, resulting in a reduction of the MVP of 0.003 vol%. This accounts for a relative volume decrease of 8% assuming an initial MVP volume of 5 vol% (Fig. 2B). Following these observations, we conclude that volatile resorption is primarily driven by recharge-induced pressurization of the magma chamber, increasing water solubility in the melt, followed by variations in phase proportions induced by increasing chamber temperatures through the addition of hotter recharge.

## The implications of volatile resorption on eruption timing
Having established that recharge-induced pressurization and decreasing crystal fraction of the resident magma can initiate volatile resorption, we assess its impact on magma chamber pressurization and stability. Therefore, we compare three volatile-resorbing cases with variable initial $H_2O$ contents (4, 4.5, and 5 wt%) with a volatile-exsolving case that has the same initial conditions as the 5 wt% $H_2O$ resorbing case but undergoes slower recharge. Looking specifically at the 5 wt% $H_2O$ containing cases, we find that pressurization rate is substantially higher in the resorbing run and that this case experiences an eruption after ~2.3 kyrs, while the exsolving run does not undergo eruption within the 5 kyrs simulation time (Fig. 3B). The elevated pressurization rates during resorption are caused not only by higher recharge rates experienced in volatile resorbing runs, but also by a

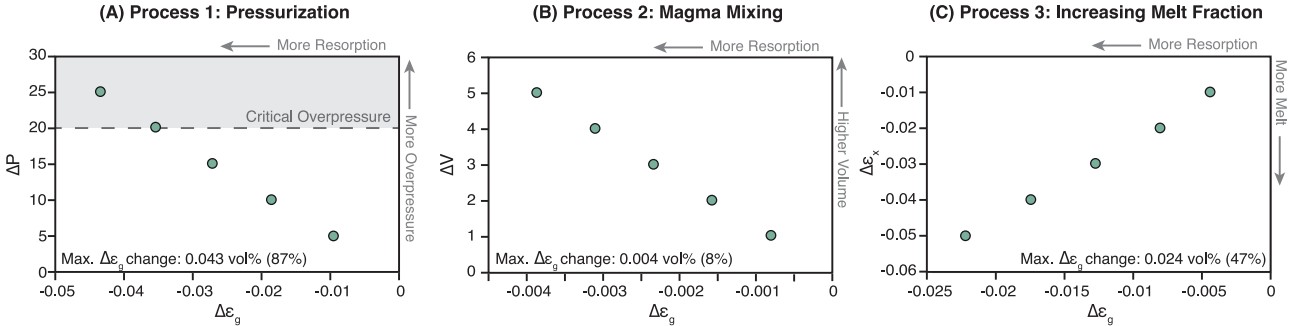

**Fig. 2 | Processes controlling volatile resorption in silicic magma chambers.** Calculated variations in MVP fractions ($\Delta\varepsilon_g$) resulting from three individual processes: **A** Process 1: recharge-induced pressurization of the magma chamber ($\Delta P$ in MPa), **B** Process 2: magma mixing of silicic resident magma with less volatile-rich recharge magmas ($\Delta V$ in km³), and **C** Process 3: changes in crystal volume fraction ($\Delta\varepsilon_x$ in vol%) induced by the dilution of resident magma with recharge and anhydrous crystal-derived melts. Changes in MVP volume fraction were calculated using the equation for water conservation in a (partially) closed system.

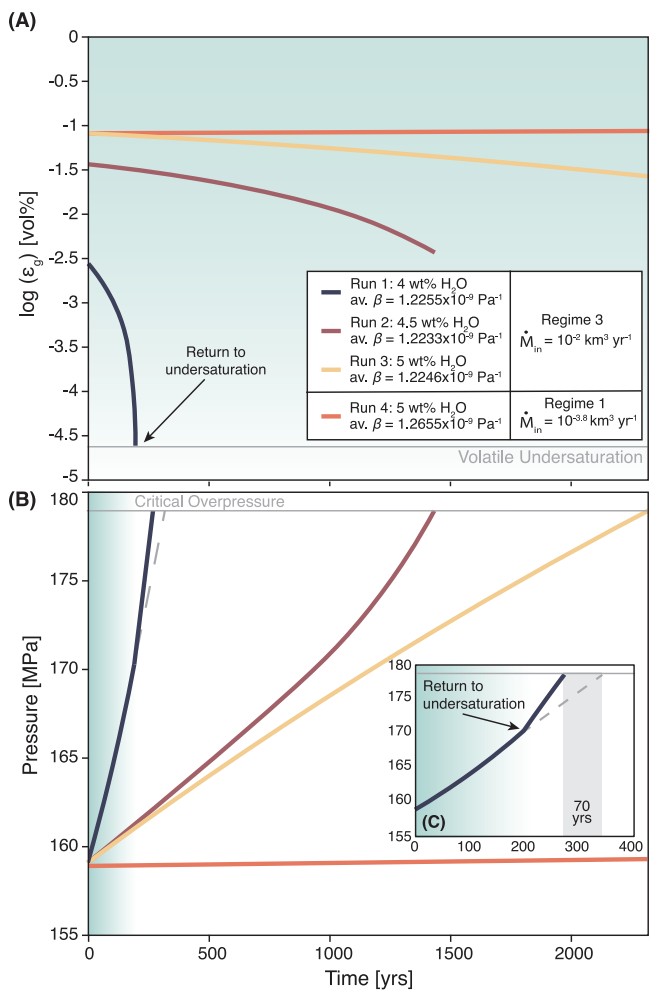

**Fig. 3 | Evolution of exsolved volatile fraction and pressure with time.** Evolution of the MVP volume fraction ($\varepsilon_g$) (**A**) and pressure (**B**) over time in Aso magma chamber simulations. Runs 1–3 undergo volatile resorption induced by elevated recharge rates ($\dot{M}_{in} = 10^{-2}$ km³/yr), while Run 4 experiences volatile exsolution under slower recharge rates ($\dot{M}_{in}$) of $10^{-4}$ km³/yr. Av. β indicates the mean magma compressibility (in Pa⁻¹) calculated after Townsend and Huber[61] averaged over the simulation time. The inset (**C**) provides a close-up of the pressure evolution of Run 1, which undergoes complete volatile resorption, transitioning from volatile-saturated to undersaturated conditions. The gray dashed line shows the extrapolated pressurization path of Run 1, reaching the critical overpressure and eruption ~70 years later than in the volatile-undersaturated case. All simulations were conducted under Aso conditions summarized in Table 1.

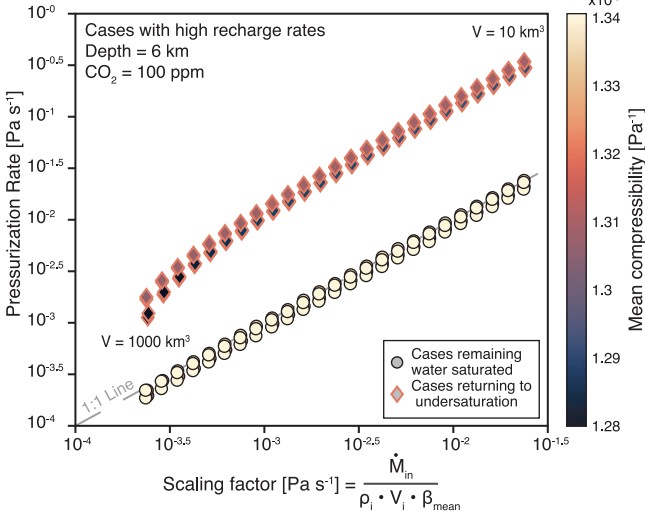

**Fig. 4 | Scaling of chamber pressurization rates.** Relationship between pressurization rate and a scaling factor, calculated from recharge rate ($\dot{M}_{in}$), initial magma density ($\rho_i$), chamber volume ($V_i$), and the mean magma compressibility ($\beta_{mean}$). Simulations ($n = 200$) were run at a constant magma chamber depth of 6 km, 2–15% initial crystallinity, initial $H_2O$ content of 3–5 wt% (in 0.5 wt% intervals), initial $CO_2$ at 100 ppm, and chamber volumes between 10 km³ to 1000 km³ (in 25 logarithmically spaced steps). Points are color-coded for the mean magma compressibility (Pa⁻¹) calculated after Townsend and Huber[61]. Diamond-shaped data points with thick pink rims indicate cases that transitioned from volatile-saturated to volatile-undersaturated conditions during simulation. Data falling on the gray dashed 1:1 line are well characterized by our scaling relation, while points above the line experience elevated pressurization.

reduction in the magma compressibility stemming from the resorption-induced decrease in MVP that usually buffers pressure build up in silicic systems (Fig. 3A). In runs transitioning from volatile-saturated to undersaturated conditions, chamber pressurization further accelerates once the system transitions to undersaturation (Fig. 3C). To fully understand the correlation between the observed lower magma compressibility and enhanced pressurization in resorbing cases, we establish a scaling relationship capturing these effects across varying magma chamber conditions (Fig. 4). The alignment of our data along the 1:1 line confirms that the scaling relationship defined by the combined effects of mass recharge rate ($\dot{M}_{in}$), initial magma density ($\rho_i$), initial chamber volume ($V_i$) and mean magma compressibility ($\beta_{mean}$) captures the first order controls on chamber pressurization. Cases that undergo a transition from volatile saturation to

undersaturation systematically fall above the 1:1-line, indicating amplified pressurization. These pressurization dynamics can result from an additional reduction of the bulk magma compressibility as the MVP volume decreases during a return to volatile-undersaturated conditions (Fig. 3A, Supplementary Fig. 3). In Aso-4, such accelerated pressurization could have led to earlier eruption onset of ~70 years compared to scenarios retaining the MVP (Fig. 3C, Supplementary Fig. 3).

## A generalized model for volatile resorption

In conclusion, we test the viability of volatile resorption as a naturally occurring process through thermo-mechanical modeling over a broad parameter space (Table 1). Focusing on the Aso-4 eruption as a case study, we confirm that pre-eruptive volatile resorption and subsequent eruption occurred if the magma chamber was recharged at rates of at least $10^{-2.4}$ km³/yr (Fig. 1C). While such recharge rates exceed typical long-term silicic magma production rates (~$10^{-3}$ km³/yr), they are consistent with previous estimates suggesting that high recharge rates on the order of $10^{-2}$ km³/yr (which are long-term averages) are required to build large silicic magma bodies with volumes over 500 km³ [5,34–37]. We further find that a return to volatile undersaturation prior to Aso-4 was possible if the resident magma contained ~4−4.5 wt% $H_2O$ and ~100 ppm $CO_2$ (Fig. 1B). This is in good agreement with geochemical estimates of the magma water content in Aso-4 magmas ranging from 4.4 to 4.7 wt% $H_2O$. We conclude that the occurrence of volatile resorption is inherently linked to high recharge rates, where chamber pressurization along with variations in phase proportions upon recharge are primary drivers for volatile resorption (Fig. 2). Magma chamber pressurization is then controlled by the combined effects of recharge rate, initial magma chamber volume and the bulk compressibility of the resident magma (Fig. 4). We further find that volatile resorption during strong recharge events generates overpressure in large-scale silicic magma chambers more efficiently than volatile exsolution. This enhanced pressurization results from both the addition of mass through recharge and the reduction of the MVP, which usually buffers the build-up of overpressure (Fig. 3). Once a system transitions from volatile-saturated to undersaturated conditions, an additional drop in magma bulk compressibility amplifies pressurization and expedites chamber destabilization leading to earlier eruption (Fig. 3C). Such an early eruption onset has significant implications for hazard assessment and risk mitigation on a human timescale and needs to be accounted for to improve forecasting of eruptive events in potentially highly hazardous systems.

If volatile resorption is a ubiquitous process in silicic systems, it would fundamentally transform current models of large-scale eruption triggering. To evaluate its prevalence, we suggest assessing water saturation levels via coupled apatite-melt inclusion analyses in systems comparable to Aso. Systems of interest should experience elevated magma flux prior to eruption, which is often considered an eruption trigger in large silicic systems [2,6–8], and is documented in cases such as the Kos Plateau Tuff [38], Bishop Tuff [39], and Bandelier Tuff [40]. In such cases, volatile resorption may occur due to increased magma flux without fully undersaturating the system, while still promoting more efficient overpressure build-up necessary for caldera eruption onset. To fully resorb the MVP and undergo further amplified pressurization, magma chambers must additionally sit near volatile saturation. Both the Bishop Tuff (3.2−6 wt% $H_2O$, saturation ~5.6 wt% [41–45]), and the Kos Plateau Tuff (4.5−6.5 wt% $H_2O$, saturation ~5.9 wt% [43,46]) fulfill this requirement, making them ideal candidates to test for volatile resorption in the lead up to cataclysmic eruptions. We find additional evidence for volatile resorption at Campi Flegrei, which exhibits a distinct recharge-driven increase in magma storage temperatures and a decrease in dissolved $H_2O$ prior to caldera-forming events. Here, the last known eruptions preceding both the Campanian Ignimbrite and the Neapolitan Yellow Tuff record $H_2O$ above saturation (5.5−6.5 wt%)

that dropped below saturation before the caldera-forming eruptions (3−6 wt%, saturation ~5.5 wt% [43,47]), suggesting potential volatile resorption.

We hypothesize that volatile resorption could be recorded in volcano monitoring through variations in the mass flux and composition of gas emissions prior to eruption. Pre-eruptive declines in gas emissions have previously been observed in silicic systems and are usually attributed to the high viscosities of silicic magmas that hinder the escape of exsolved volatiles or progressive sealing of the conduit (e.g. refs. [48,49]). Volatile resorption, however, could have similar emission-reducing effects that were previously unrecognized. Similarly, the $CO_2/SO_2$ ratio of gases emitted prior to eruption could be impacted by resorption. Since $SO_2$ is more soluble in silicic melts than $CO_2$, we expect an increase in the $CO_2/SO_2$ ratio as a response to beginning volatile resorption during continuous recharge [50]. Such dynamics cannot be captured with our model, which does not solve for $SO_2$, but are frequently observed prior to eruption, being primarily linked to fresh recharge entering the system (e.g. refs. [51,52]). Ground deformation and seismicity are expected to continue or even intensify during volatile resorption. Kilbride et al. [53], suggest that ground deformation can be muted by the presence of a compressible MVP. Likewise, we expect a resorption-induced reduction of the MVP to cause a decrease in magma compressibility, while resorption of bubbles would lead to a decrease in the magma chamber volume. Such competing effects could fluctuate or enhance ground deformation prior to large silicic eruptions, as is periodically observed at Campi Flegrei [54]. For example, during the 1982-84 unrest an uplift of ~1.9 m was observed accompanied by a sharp increase of the $CO_2/H_2O$ ratio, which could be caused by resorption [55,56].

We propose that volatile resorption is a critical, yet potentially overlooked process that significantly impacts reservoir pressurization and stability even without transitioning a chamber to undersaturation. While our framework establishes the fundamental viability of this process, we recognize that significant uncertainties remain when translating a numerical model to complex natural magmatic systems. Therefore, this work is intended to provide a foundation for future research. Specifically, the integration of more complex physical parameters, such a magma chamber geometry or the influence of variations in tensile strength of host rocks would be essential to capture the complexities of volatile resorption in diverse volcanic settings. Identifying volcanic systems that underwent volatile resorption in the past and monitoring large-scale silicic volcanoes for it could help us identify highly recharged systems that undergo amplified pressurization with the potential of near-future eruptions.

## Methods

To test the physical and chemical viability of volatile resorption and simulate the thermodynamic and mechanical response of magma chambers to it, we apply the numerical framework developed by Scholz et al. [21]. This framework is based on a 1D box model introduced by Degruyter and Huber [2] and further developed by Townsend et al. [3], to investigate the thermo-mechanical evolution of three-phase silicic magma chambers in a visco-elastic upper crustal setting. Magma chambers are defined as the mobile portion of a magmatic reservoir, containing less than 50 vol% crystals, silicate melt and if saturation is reached, a magmatic volatile phase (MVP). During simulation, the chamber undergoes mass injection via recharge and mass loss through episodic eruption, while experiencing conductive heat loss to the surrounding wall rocks. The updated model by Scholz et al. [21], expands upon the initial water-only setup by incorporating $H_2O$-$CO_2$ fluid mixtures and allowing for the simulation of both mafic and silicic compositions. The solubility of $H_2O$ and $CO_2$ is described using the parametrization from Liu et al. [43] for rhyolitic melt under the assumption that volatile partitioning between the melt and MVP takes place under chemical equilibrium conditions. To test the kinetic feasibility of

volatile resorption, we calculate water diffusivity for Aso-Y compositions at 900 °C and 200 MPa after Ni and Zhang[57]. With a water diffusivity of $3.643 \times 10^{-10}$ m²/s, complete MVP resorption would take ~0.1–1100 s for bubble sizes between 10 and 1000 μm in a 500 km³ magma chamber containing 4.5 wt% initial $H_2O$. Throughout the simulation, phase proportions of melt, crystals and MVP vary with changes in magma chamber pressure, temperature, and volatile solubility in response to continuous magma recharge and conductive heat loss. We use the equations for the conservation of total mass, $H_2O$ mass, $CO_2$ mass, and enthalpy to track variations in magma chamber mass, pressure and volume over time. Mass removal from the magma chamber occurs via eruptions, which are initiated once a critical overpressure (i.e. magma pressure in excess of lithostatic pressure) is reached. We assume the presence of pre-existing weak zones such as fractures and faults around the magma chamber; therefore, magma overpressure needs to be high enough to reactivate weak zones and propagate a dike faster than it freezes. In the literature, such overpressure threshold values vary between 10 and 40 MPa dependent on host rock properties and magma viscosity. Here, we use an overpressure threshold for fracture re-activation and eruption onset of 20 MPa. Eruptions remove the required amount of magma mass to restore the magma chamber pressure to lithostatic.

## Calibration of the model to Aso compositions

To get a clear understanding of the magma dynamic processes preceding the Aso-4 event, we calibrated the model presented by Scholz et al.[21], to Aso-specific compositions. We selected a rhyodacitic melt composition measured from glass shards of the volatile-saturated Aso-Y eruption[22] that occurred ~5 kyrs prior to the Aso-4 event at ~91.4 ka (Supplementary Table 1). To calibrate our model, we used a Matlab-based batch processing tool of the rhyolite-MELTS software[58], generating temperature-crystallinity curves for Aso-Y as model input. These curves were defined across a broad range of pressures (50–400 MPa in 25 MPa intervals), temperatures (700–1200 °C in 1 °C intervals), $H_2O$ (0.5–5 wt% in 0.25 wt% intervals) and $CO_2$ contents (0–500 ppm in 50 ppm intervals with additional runs at 1 and 10 ppm). MELTS runs were conducted under isobaric equilibrium crystallization conditions at an oxygen fugacity of NNO + 2.65[59]. The resulting mass fractions for crystals, melt and fluid were validated for mass-balance and converted to volume fractions for further processing.

To describe crystallinity as a function of temperature, pressure, and $H_2O$ content in our model, we fit the MELTS outputs following the method described in Scholz et al.[21]. They found that temperature-crystallinity curves of rhyolites are best fitted using error functions, while average basaltic compositions can be described using linear functions. To fit the rhyodacitic Aso-Y compositions, we therefore apply a segmented regression that combines linear and error functions with a smooth transition, covering a crystallinity range from 3 vol% ($\varepsilon_x = 0.03$) to 80 vol% ($\varepsilon_x = 0.8$). We find that low crystallinity segments of the curve are best described using a linear fit following:

$$\varepsilon_x = A \times T + B, \tag{1}$$

while higher crystallinity segments are best described with an error function:

$$\varepsilon_x = C \times \mathrm{erfc}(D \times (T - E)) + F. \tag{2}$$

$\varepsilon_x$ stands for the crystal volume fraction, $T$ for the temperature of the run (°C), and $A$, $B$, $C$, $D$, $E$, and $F$ are fitting parameters. Each fitting parameter is calculated individually as a function of $H_2O$ content ($\chi_t^w$ in wt%) and pressure ($P$ in MPa) as described in the following equations. Since the effect of $CO_2$ on parameter fits is small compared to the impacts of $H_2O$, we neglected it here. $T_0$ describes the transition point

between the two curve fitting equations.

$$A(\chi_t^w, P) = \frac{F(\chi_t^w, P) - B(\chi_t^w, P) + C(\chi_t^w, P) \times \mathrm{erfc}(D(\chi_t^w, P) \times (T_0(\chi_t^w, P) - E(\chi_t^w, P)))}{T_0(\chi_t^w, P)} \tag{3}$$

$$\begin{aligned} B(\chi_t^w, P) = {} & 3.406 - 1.298\chi_t^w + 6.78 \times 10^{-5}P + 0.3318\chi_t^{w2} + 0.0001133\chi_t^w P \\ & - 3.387 \times 10^{-8}P^2 - 0.02805\chi_t^{w3} - 6.757 \times 10^{-6}\chi_t^{w2}P \\ & - 2.393 \times 10^{-8}\chi_t^w P^2 + 1.203 \times 10^{-11}P^3 \end{aligned} \tag{4}$$

$$\begin{aligned} C(\chi_t^w, P) = {} & 0.2039 - 0.03967\chi_t^w + 2.62 \times 10^{-5}P + 0.006702\chi_t^{w2} + 2.523 \times 10^{-5}\chi_t^w P \\ & - 3.251 \times 10^{-8}P^2 - 0.0006055\chi_t^{w3} - 3.335 \times 10^{-6}\chi_t^{w2}P + 4.296 \times 10^{-9}\chi_t^w P^2 \end{aligned} \tag{5}$$

$$\begin{aligned} D(\chi_t^w, P) = {} & 0.05491 - 0.02003\chi_t^w + 5.98 \times 10^{-5}P + 0.01834\chi_t^{w2} - 1.544 \times 10^{-5}\chi_t^w P \\ & - 1.275 \times 10^{-8}P^2 - 0.002331\chi_t^{w3} + 7.214 \times 10^{-7}\chi_t^{w2}P + 2.786 \times 10^{-9}\chi_t^w P^2 \end{aligned} \tag{6}$$

$$\begin{aligned} E(\chi_t^w, P) = {} & 856.6 - 7.875\chi_t^w - 0.09838P + 1.17\chi_t^{w2} + 0.0006271\chi_t^w P + 3.006 \times 10^{-5}P^2 \\ & - 0.0001973\chi_t^{w2}P + 5.233 \times 10^{-8}\chi_t^w P^2 - 3.4 \times 10^{-9}P^3 \end{aligned} \tag{7}$$

$$\begin{aligned} F(\chi_t^w, P) = {} & 0.7868 - 0.4093\chi_t^w + 0.0002331P + 0.1121\chi_t^{w2} - 4.308 \times 10^{-5}\chi_t^w P \\ & - 2.535 \times 10^{-8}P^2 - 0.009755\chi_t^{w3} + 9.498 \times 10^{-8}\chi_t^{w2}P + 5.113 \times 10^{-9}\chi_t^w P^2 \end{aligned} \tag{8}$$

$$\begin{aligned} T_0(\chi_t^w, P) = {} & 868.5 - 3.51\chi_t^w - 0.09669P + 0.0008189\chi_t^w P + 2.766 \times 10^{-5}P^2 \\ & - 1.037 \times 10^{-7}\chi_t^w P^2 - 2.959 \times 10^{-9}P^3 \end{aligned} \tag{9}$$

A comparison between crystallinity-temperature curves generated via MELTS and our parametrization using the presented equations can be found in Supplementary Fig. 2. The deviations between the original MELTS crystal volume fractions and the calculated fits range from 0 to 10 % for all $H_2O$ and $CO_2$ contents (Supplementary Fig. 2).

## Parameter space

A total of over 12,000 simulations were run to analyze magmatic processes influencing the dynamic behavior of volatiles and their potential impacts on eruption onset. In a first step, we performed a broad parameter search covering four orders of magnitude of magma chamber volumes (10–1000 km³) at crustal depth of 6 and 8 km. To simulate variable starting temperatures, we apply initial magma crystallinities of 2 and 15 vol%. We use a fixed and prescribed starting crystallinity instead of a starting temperature since liquidus temperatures can vary considerably with the total volatile content of the magma (starting temperatures for the 15 vol% crystals vary between ~855 and 900 °C). Initial $H_2O$ contents covered a wide range between 3 and 5 wt%, while initial $CO_2$ contents were set to 100 and 500 ppm to represent high and low $CO_2$ endmember scenarios. These volatile contents represent total volatile contents within the system, which include both volatiles dissolved in the melt and exsolved into the MVP. We additionally tested the impact of variable recharge conditions on volatile dynamics by varying recharge rates from $10^{-2}$ km³/yr to $10^{-4}$ km³/yr (corresponding to 7.1–713.5 kg/s for a melt density of 2250 kg/m³ calculated for Aso-Y melts) across 100 logarithmically spaced steps, adjusting recharge temperatures to 900 °C and 1300 °C, and setting recharge volatile contents to 1 and 4 wt% $H_2O$ and 100, 500, 1000, and 10,000 ppm $CO_2$. Due to model constraints, the composition of the

recharging magma is identical to the resident magma, however through variations in temperature and volatile enrichment, we model less evolved recharge conditions, as recorded before the Aso-4 event[25,60]. Simulations were then run until one of the following conditions were achieved: (1) a crystallinity of 50 vol%, marking rheological locking, (2) a crystallinity of 0.01 vol%, preventing complete remelting of crystals, or (3) a maximum run time of 5 kyrs, representing the time interval between the Aso-Y and Aso-4 events. In an additional step following the wide range parameter search, we constrained magma chamber volumes to 500 km$^3$[24] and chamber depth to 6 km (e.g. refs. [28,29]) to match Aso-specific conditions. An overview of the full and Aso-specific parameter spaces is provided in Table 1.

## Data availability
The source data used to generate the display items are provided as Source Data files. Run files required to reproduce the simulation data are included with the accompanying code (see Code Availability Statement). Source data are provided with this paper.

## Code availability
All MATLAB scripts used to perform the simulations and generate figures are archived on Zenodo at https://doi.org/10.5281/zenodo.15609585. The package includes run files required to reproduce the results presented in this study.

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

## Acknowledgements

F.K. acknowledges support from the Swiss National Science Foundation (SNSF) through a Postdoc.Mobility Fellowship (Grant No. 214248). M.T. and C.H. acknowledge funding from the U.S. National Science Foundation (NSF), Division of Earth Sciences, under Grant Nos. 2444709 and 2121655, respectively.

## Author contributions

F.K. performed the formal analysis, wrote the original manuscript draft, and conceptualized the study in collaboration with M.T. M.T. supervised the project and directed the overarching study design, including the technical implementation of the numerical setup and the conduction of simulations. J.T. provided expertise in Rhyolite-MELTS modeling and the calibration of essential Aso-specific melting curves. C.H. contributed to the development of the modeling approaches and formulated the calculations related to resorption mechanisms. All authors discussed the results, engaged in data interpretation, and contributed to the critical review and finalization of the manuscript.

## Competing interests

The authors declare no competing interests.
