## [Transparent Peer Review file · Nature Communications]

Volatile resorption expedites eruption onset in large silicic systems

Corresponding Author: Dr Franziska Keller

Version 0:

Reviewer comments:

Reviewer #1

(Remarks to the Author)

This is an interesting paper that explores the role of pressurisation of a magma reservoir upon recharge. Using an established code (ie an already published one by the same group) the authors explore how the excess fluid/gas that a reservoir may hold prior to recharge may be resorbed in response to the pressure increase. The authors show that under some conditions of magma recharge, that excess fluid may be fully redissolved in the melt, which results in a significant decrease of its bulk compressibility, hence in an increase of the rate of pressure increase, which in turn may bring the reservoir to its failure at a faster rate as well. It is true that the conventional wisdom in this area is to consider that cooling increases excess fluid amount hence magma eruptability. This study shows that this may not always be the case and on that ground it provides an interesting and novel view on this complex problem. This study is a logical follow up of previous efforts led by the same team (Keller et al., 2013 in EPSL) in which the authors proposed the idea of a shift between gas saturated and under-saturated conditions of the Aso reservoir (using apatite volatiles systematic), and suggested that this could affect eruption frequency. In the present contribution they carry out the numerical simulations testing their hypothesis, which they find is a physically tenable one.

Being not a numerical modeller I can only trust what the model predicts, but the core code has been already used in a number of published papers, so I assume it has been already benchmarked and it is working properly.

My comments are mostly suggestions, clarifications asked for, and some are possibly naive questions given my limited experience in numerical simulations:

1. The authors investigated a range of volatile contents, either for the resident or the intruded magma. For the later, it is my understanding that it aims at simulating conditions relevant to a mafic recharge (as inferred for Aso by previous work, in particular Kaneko et al., 2007). My question is why the bulk CO₂ of that mafic end-member is restricted to 1000 ppm or less? There is evidence that the CO₂ content of mafic magmas, including in arcs, can reach several wt% (see the case of Stromboli for instance). It would be interesting to expand the CO₂ range to at least 1 wt% bulk content (as in Scholz et al., 2023) and see how this could affect model's outcome.

2. I also guess or understand that the role of recharge is to provide heat and mass (including H₂O and CO₂?) to the reservoir but only heat is allowed to propagate upwards (in addition to that lost to the host rock)? In other words the melt fraction trends which are shown in extended data Figure 2 are those which are used to simulate the bulk of the magma reservoir? Or do you take into account also the mafic layer? Similarly, are the H₂O and CO₂ sourced from the mafic recharge transferred into the overlying silicic melt? I would also suggest the authors to compare their curves with experimental data to assess how accurate is MELTs in calculating melt fraction at a given bulk H₂O and CO₂ content: for instance the phase equilibria of Costa et al (2004) have been performed on a whole rock composition very similar to that selected in the present work (see also Scaillet and Evans 1999, though this uses a slightly more mafic magma).

3. From what I understand, the compressibility of host rocks β is kept constant in all simulations. Presumably this property is likely to vary between volcanic areas, depending on the level of intrusion, how fractured are the host rocks etc...The authors fix compressibility to $\beta=1/10-10$ Pa (correct?). Quoting Gudmundsson (Volcanotectonics, 2020, page 286) it is said that host

rock compressibility can be $\beta=3 \times 10^{-11} \text{ Pa}^{-1}$. I have no opinion on which value is the most appropriate but a comment of why the value $1/10^{-10} \text{ Pa}$ has been selected would be welcome. I am asking this because using my own rudimentary code which simulates the evolution of internal pressure of a cooling magma reservoir, made me understand that the host rock bulk modulus is quite important in determining how much of a pressure increase arises (and not only the magma compressibility).

4. One of the great appeal of the model, for a non expert like me, is that it allows to simulate the time evolution of the reservoir, which is a highly desirable property for any model aimed at anticipating eruptions with some confidence. However, the age of Aso-4 event seems to come from Hbl Ar dating (Albert et al., 2019 quoted by Keller et al., 2023), but that phase is described as being xenocrystic or not at equilibrium with its host melt (Ishibashi et al., 2018) which is supported by the lack of Hbl in the run products of hydrothermal experiments of Ushioda et al (2020). I wonder therefore how sure one can be about the time interval between Aso-Y and Aso-4? Is there any other data or way to ascertain the age of Aso-4?

5. One important result or at least, petrologically speaking, the one that brings question to my mind is that the reservoir may undergo a significant pressure increase during cooling (as in previous work). Is this pressure increase documented (aside from apatite volatile record)? I mean, in an ideal world, one could expect to see phenocrysts hosting melt inclusions showing such a pressure increase, ie an increase of both CO₂ and H₂O contents as MI get closer to the rim? I know this is difficult task (in particular in phenocryst-poor rocks and given the rather modest changes in pressure) but in my opinion it would be the real smoking gun of the process advocated here. Also, if I understood correctly, volatile resorption comes mostly from an increase in pressure but not from a decrease in crystal fraction? It would help the reader to show a graph with the evolution of these two parameters (gas and crystal fractions) with time during a single but representative simulation that culminates in gas-free conditions close to reservoir rupture. Increasing pressure while cooling and preserving no textural evidence of melting/crystallisation requires quite a subtle combination of parameters. According to petrological constraints (Keller et al., 2023, JVGR), the temperature drop between Aso-A and Aso-4 is about 55°C. Does that fit with the model output?

6. The model considers the EOS derived by Halbach and Chatterjee on H₂O to simulate the PVT of the gas phase. This EOS is for H₂O only and it seems that it is used to compute the volume properties of the H₂O-CO₂ fluid mixture. Am I correct? If so, any idea of how a more real EOS (fluid mixture) would affect the results? More specifically, I don't understand how equilibrium distribution of both H₂O and CO₂ species between gas and melt can be computed with an EOS for water only?

7. I think that Bishop Tuff may not be a good example of a system sitting near gas saturation. According to Wallace et al (1995,1999) seminal work, this system records good evidence of being already gas saturated. As for Campi Flegrei, I don't believe we have the firm evidence in hand for such a shift toward gas under-saturation, since temperature estimates of that system are not that well constrained in my opinion.

8. Pre-eruptive declines in gas emissions also arise from progressive sealing of the volcanic conduit, and not only because of the high viscosity of the magma

9. I don't think it is correct to say that CO₂ solubility is higher in mafic melts relative to more silicic ones (line 186). CO₂ solubilities are not that much different in mafic and felsic melts, and in fact, a rhyolite could dissolve even more CO₂ if held at the same fCO₂ and T than a basalt liquid. What happens in nature is that by the time silicic melts are produced most of the CO₂ of the system has been lost.

10. From what you have done, I think that the best pre-eruptive signal of a pressurising system entering a gas-free regime would be a break in slope of geodetic signals recording ground uplift (following what is shown in your figure 3b(inset)), assuming the time series of data is long enough to capture this change? Yet, the Campi Flegrei example shows that the rapid up and down ground motions are not that easy to interpret in terms of deep seated processes..

Bruno Scaillet.

(Remarks on code availability)

The code has been used several times by the same team, and has led to several publications which I assume have permitted to check it

Reviewer #2

(Remarks to the Author)

Review of "Volatile resorption expedites eruption onset in large silicic systems" by Keller, Townsend, Troch, and Huber.

I have read this manuscript closely and with great interest. The authors use the case study of the caldera-forming Aso-4 eruption to investigate the hypothesis that mafic recharge of large silicic magma chambers can cause resorption of exsolved magmatic volatiles, and that this resorption can then cause increased chamber pressurization rates through the corresponding decrease in magma compressibility, resulting in earlier eruption onset. The authors use thermal mechanical modelling in conjunction with previously published geochemical data regarding the pre-eruptive chemistry, volatile contents and magma chamber conditions of the Aso-4 eruption. Their results indicate that when recharge rates are high volatile resorption can occur, primarily through chamber pressurization (which increases volatile solubility in the melt) and increase of the melt volume fraction by addition of hot recharge melts that also act to melt existing crystals in the resident magma (which together adds relatively volatile-poor melt thus reducing the average volatile content of the melt, approaching a state

of volatile undersaturation). The authors suggest that elevated pressurization rates caused by complete volatile resorption could have brought forward the onset of the Aso-4 eruption by ~70 years. The authors propose that detecting precursory signals of chamber pressurization caused by volatile resorption could be critical for early warning of future eruptions.

Overall I am convinced that volatile resorption may be a common (or, not uncommon) process in large silicic magma chambers and that a consequence of the resorption process can be increased rates of chamber pressurization that may trigger eruption. I agree it is an important topic and worthy of future study. However, I am left uncertain about the extent of the authors' argument regarding volatile resorption as an eruption trigger, and unconvinced by how useful or meaningful it is that the Aso-4 eruption may have been 'expedited' by up to 100 years. This latter point is a significant issue for a paper with the current manuscript title.

The paper is generally well-written and logically set out, however I was still left with many questions about the authors' precise meaning. I have included these in my comments below and I expect that most of them can be addressed by tighter and more explicit text, particularly for figure captions, and perhaps some minor adjustments to the figures. Addressing them will avoid unnecessary confusion for the reader, particularly for the manuscript layout that provides few figures in the main text with detailed methods at the end. The most important revisions will be those addressing to what extent and why the authors are arguing for the importance of volatile resorption in understanding large silicic eruptions. If these can be addressed convincingly I would recommend the manuscript for publication in Nature Communications.

I have listed my detailed comments below. Line numbers refer to the line numbers of the PDF version provided for review.

Title

'expedites eruption onset'. Is this the most important finding? The slightly earlier eruption?

Abstract

L16 What is the meaning of 'independent' here? Previously published?

L16 and throughout: kyrs is a length of time. An eruption 86 thousand years ago should be 86 ka or 86 kyrs BP, etc

Main text

L36-56 this is a nice (clear, concise) explanation of the study context

L52 are vapor loss by passive degassing/viscous relaxation of the crust included in the magma chamber model used? The model description includes 'in a visco-elastic crust' (L76), what about passive degassing?

L55-56 "The high compressibility of the MVP increases bulk magma compressibility, dampening pressurization from recharge and leading to less frequent, yet larger, eruptions". Reading this, I wonder if the proposed volatile resorption mechanism should have any expected effect on 1) frequency and 2) size of eruptions? Certainly Aso-4 was caldera-forming even though undersaturated... would it have been even bigger if the initial magma was not undersaturated?

L61 another example where kyrs should be ka or kyrs BP etc

L64 water undersaturated. Was CO₂ saturated i.e. CO₂ bubbles present in the initial magma? Or the argument is the eruption could start without any MVP at all?

L64 why is passive degassing of the MVP ruled out? If the answer is in the other Keller et al paper please add it here concisely for the reader

L66 kyrs is used correctly here

Volatile dynamics prior to the Aso-4 caldera-forming eruption

See L52 comment above, does model include passive degassing effects

L81-82 recharge magma is 'drier and hotter'. Is it mafic? Is the recharge melt composition included in the magma mixing modelling? And is it accurate to call it 'drier' or is 'less volatile-rich' better – what about CO₂ as well as H₂O?

Figure 1 caption is confusing. Do all three panels show data with the identical initial parameters as in the first sentence of the caption? Are the data in B and C for the same runs, i.e. this figure shows the results of varying resident magma H₂O wt% and recharge rate and nothing else? It took me some detailed re-reading to reach this (hopefully correct) understanding. Panels B and C easily look to a quick glance like they are plotting temporal variations in crystallinity & MVP during three runs, with start point in upper right and end point in lower left. Are the symbols showing the final change in crystallinity & MVP at the end of each run? Why are some 'undersaturated' diamonds shown within Regime 2 – are these runs where the MVP totally resorbed but then some volatiles exsolved again before the model run ended?

In general it is better to have a lengthy caption to prevent unnecessary confusion for readers seeing this diagram for the first time. I am not a fan of 'show don't tell' in figure captions – better to 'show and tell, then let the reader decide if they agree'.

Related, Extended Data Figure 1 was extremely helpful in understanding this manuscript and the meaning of Fig. 1 and it is unfortunate it is not in the main manuscript. Could it not be combined to make the original Fig. 1 a 6 panel figure?

L93 “we define four distinct regimes that describe the coupled thermal and volatile saturation conditions in the chamber as a function of recharge rate (Fig. 1A)”. There is nothing in Panel A to indicate the influence of recharge rate on these regimes, and the recharge rate is argued to be a crucial factor in the volatile resorption mechanism. Can useful schematic arrow(s) of ‘increasing recharge rate’ be applied to this regime illustration? Or again, it is an argument to combine the Extended Data Figure 1 here.

L98 Regime 4 is not observed for the Aso data, why not? What conditions would need to be met? Is it possible at other volcanic systems or is it an unnatural scenario?

L108 most(?) of the model runs that resulted in eruption within the 5 kyr model timeframe were not fully undersaturated (i.e. not diamond symbols). Why is it so important that Aso-4 was back to undersaturated conditions? What difference (eruption size, duration?) between the runs that were undersaturated vs partially resorbed but still with MVP? Is the authors' argument that magma chambers that enter Regime 3 will have eruptions sooner than those that stay in Regimes 1-2 (or 4?), even if the MVP is not fully resorbed? Or is full MVP resorption considered to cause the most rapid return to eruption?

What drives volatile resorption during magma recharge?

Processes 2 and 3 need clarification, between the description in the text here and the caption of Figure 2.

What are the 5 data points shown in Figure 2? Is the change in MVP vol% by these difference processes being output from model runs like those in Figure 1, or just a simple calculation of ‘suppose Process X causes an overall change in parameter Y of amount Z’?

Process 2:

Line 111 “(2) mixing of silicic resident magma with drier and hotter recharge magma”, but also

Line 128 “The effect of chemical mixing between the resident and recharge magmas (2)”, and also

Fig 2B caption “(B) magma mixing of silicic resident magma with drier recharge magmas (Solubility in wt%)” (Line 461).

What is being included in the calculation for Process 2, as shown in Fig 2B as ‘delta solubility wt%’?

Change in solubility value caused by change in SiO₂ content of resident magma by mixing of initial rhyodacite with mafic recharge? (Does this study's model include the chemistry of the magma? If not, why not – if it is shown to be negligible, say so).

And/or the change in solubility value caused by temperature increase due to addition of hotter recharge magma?

As for ‘drier’, solubility value of a volatile in a given melt composition at a given P,T condition is fixed. Mixing one dry (volatile-poor) version of a melt into a wet (volatile-rich) version of that melt does not change the solubility value. So it seems ‘drier’ should be deleted from the description of Process 2.

Instead of ‘delta solubility’, are the authors aiming for a meaning of ‘closer to/further from saturated conditions’? This would more accurately reflect the Process 2 mixing described in Line 129-130 (unless, there is an unspoken assumption about different temperature and composition of the recharge vs resident magmas).

Process 3:

Line 112 “(3) variations in phase proportions of crystals, melt and MVP through influx of less crystalline recharge magma”

Fig 2C caption “(C) changes in crystal volume fraction (x) induced by the dilution of resident magma with anhydrous recharge and crystal-derived melts.” (Line 462)

It seems heat (hotter recharge magma) is missing from these outlines of Process 3, although the effect of heat for melting crystals and liberating anhydrous melt is indeed described in the following text.

Why doesn't Fig 2B have change in gas fraction going from 0 on the x axis, like panels A and C? Why does ‘reduced solubility’ (as indicated by the y axis) still correspond to decreases in gas fraction – lower solubility should mean more of the volatiles in the system should be exsolved in bubbles?

The implications of volatile resorption on eruption timing

Fig. 3 explain the notation in the panels in the figure caption.

Line 468 ‘inset’ typo

The three eruptions in Fig 3 all happen on much shorter timescales than the 5 kyrs between Aso-Y and Aso-4 that is(?) being modelled. Are these scenarios realistic?

Line 147-148 The finding that Aso-4 eruption occurs 70 years earlier when the MVP is totally resorbed rather than partially resorbed (but still in Regime 3) seems to be the big crux of this paper. But with timescales of thousands of years since the

last eruption, does this plus/minus 70 years have any practical value? We are not able to make eruption predictions at anything like that level of precision (if at all).

Line 162-164 “We find that once a system transitions from volatile-saturated to undersaturated conditions, a drop in magma bulk compressibility amplifies pressurization expediting chamber destabilization and earlier eruption”. Is the change to undersaturated conditions (no MVP) the critical point for ‘expediting’ eruptions? As said above, an earlier onset of 70 years does not seem such a remarkable finding. What about Regime 3 in general? If a large silicic magma chamber is experiencing high recharge rates, does it experience an earlier eruption if its initial conditions (density, size etc) cause it to evolve through the Regime 3 pathway (even if the MVP is not totally resorbed) compared to if its initial conditions led it to evolve through Regimes 1,2 or 4? This might be of greater impact in understanding which systems could erupt quickest and necessitate closest monitoring of temporal variation of precursory signals.

Line 167 again I am not sure to what extent the authors are arguing for the importance of resorption as an eruption trigger. Does this trigger require total resorption of the MVP or is only partial resorption enough? Or is their argument that these big systems (Bishop Tuff, KPT, CF) require volatile undersaturation to erupt; their high recharge rates are not enough to trigger eruption unless they also trigger undersaturation?

L187 what is the reasoning for the expected increase in CO₂/SO₂ ratio? If CO₂ becomes more soluble as the system becomes more mafic (through mixing of mafic recharge), then more CO₂ dissolves and CO₂/SO₂ ratio should instead decrease. Or CO₂ solubility is higher in mafic melts thus the mafic recharge is bringing in lots more CO₂ to the chamber hence higher CO₂/SO₂ – but how is this different to the case of mafic recharge that doesn’t trigger resorption?

L192-193 The link between the volatile resorption mechanism and the observable precursory ground deformation and perhaps seismicity signals that might result seems like a more useful and significant outcome for hazard mitigation than the finding that resorption might cause eruptions to occur a few decades earlier than they otherwise would. It needs a lot more work, probably beyond the scope of this study, to know if this is feasible.

L194 the reference for the CF deformation is for recent observations. Are the authors implying these deformations could be happening because the CF magma chamber is currently undersaturated? Do the CF CO₂/SO₂ or other observations over the same timescale support their interpretation?

L197-198 “identify highly recharged systems that undergo amplified pressurization with the potential of near-future eruptions” Again I am left wondering what this means. Systems with the potential for total MVP resorption? Or partial MVP as well? And the meaning of ‘near-future’. The three eruption scenarios in Fig 3 all take place within ~300 to 2500 years. Is this realistic? And what type and size of eruption will result? See also my comment on L55-56.

Methods

L359 do different run conditions have different erupted magma volumes? Is volatile undersaturation exclusively associated with large caldera-forming eruptions?

L363 Aso-specific compositions. The following calibration for the Aso-Y crystallinity with T, P, H₂O, does not include the effect of mixing in mafic magma with low SiO₂. Is this change in overall melt chemistry shown to be insignificant for the melting of crystals etc that promotes resorption?

L423-424 if recharge is mafic it doesn’t make sense to use the density of Aso-Y rhyodacite melt

(Remarks on code availability)

Reviewer #3

(Remarks to the Author)

Summary

The authors build upon previous observational results of mineral chemistry from the Aso-Y and Aso-4 eruptions to consider the implications of mafic recharge in large silicic systems on volatile saturation state and the consequences on eruptability. Simulations using an existing numerical model solve for the evolution of the state of the magma chamber, including pressure, temperature, and chemistry and how perturbations to these change the volume fractions of crystals and vapor which have an impact on the magma chamber compressibility. While I think the core argument that mafic recharge could induce some degree of volatile resorption is sound, I have questions about the details and the strength of the authors’ claims of magma chamber-wide undersaturation before the manuscript is ready to be published.

Distinguishing mechanisms of resorption

Overall, I like the approach taken to tease apart the multiple possible mechanisms to identify the most important. However, I think there are some issues in the calculation:

1) the assertion for constant chamber volume ($V_0 - V_i$) is, I think, causing a problem here. The simulations with volatile resorption presented in Fig. 1 clearly show a change in chamber volume. For example, the high flux and 4.5 wt% initial

water case; $\Delta\epsilon_g \approx -0.03$ and $\Delta\epsilon_x \approx -0.035$, by mass conservation of the melt $\Delta\epsilon_x \approx \Delta\epsilon_m$, $\rho_x = 2800$, $\rho_m = 2250$ and $\rho_g = 350$ then $(-0.03 \cdot 350 - 0.035 \cdot 2800 + 0.035 \cdot 2250) / (0.05 \cdot 350 + 0.15 \cdot 2800 + 0.8 \cdot 2250)$ then the chamber volume is changing by $\sim 1\%$ which could be important here given the small initial bubble volume. Also, you can see with the joint constraint of $\Delta\epsilon_g + \Delta\epsilon_m + \Delta\epsilon_x = 1$ (which is true by definition) that you are forcing the decrease in the gas volume fraction (and therefore absolute gas volume) to occur at the expense of crystal and melt volume, which is probably why you come to the conclusion that the crystal melting (and resultant volume expansion of the crystal+melt assemblage) is the important driving factor.

2) you're solving here for only the water mass balance between the bubbles and melt, but in the mixed H₂O-CO₂ system, the CO₂ really matters. If you have 4.5 wt% H₂O and 300 ppm CO₂ total, the vapor would have 30 vol% CO₂ (50 wt%). Which has a direct effect on the vapor volume and you need to consider the resorption of CO₂ to get the right vapor volume. But also, CO₂ has a strong effect on the water solubility. I ran a quick mass balance calculation (ignoring the input magma for hypothesis 1) for joint water and CO₂ starting from:

$$MH_2O = \rho_m(1 - \epsilon_g, 0 - \epsilon_x, 0)_{meq, H_2O} V_0 + \rho_{-H_2O} \chi_{-H_2O} \epsilon_g V$$

$$MCO_2 = \rho_m(1 - \epsilon_g, 0 - \epsilon_x, 0)_{meq, CO_2} V_0 + \rho_{-CO_2} (1 - \chi_{-H_2O}) \epsilon_g V$$

Where χ_{-H_2O} is the mass fraction of water in the bubble. And from a little rearrangement, assuming no change in the melt and crystal volumes: $V = [1 - \epsilon_g, 0 + \epsilon_g(1 - \epsilon_g, 0)] / (1 - \epsilon_g)] V_0$. I don't have your exact numbers, but if I choose a pressurization from 200 MPa to 220 MPa along with a temperature increase from 875 °C to 913 °C (for addition of melt at 1300 °C at a rate of 10-2 km³/yr, although the result is similar in the iso-thermal case) and 15% crystallinity along with $\rho_{-H_2O, 0} = 381$ kg/m³, $\rho_{-H_2O, i} = 389$ kg/m³, $\rho_{-CO_2, 0} = 922$ kg/m³, $\rho_{-CO_2, i} = 982$ kg/m³ using the ideal gas law for CO₂ and the Pitzer and Sterner (1994) equation of state for water and the Liu et al. (2005) mixed solubility model, I find that a starting vesicularity of 0.05 and starting melt composition of 4.55 wt% H₂O and 370 ppm CO₂ (equilibrium for 4.5 wt% H₂O total and 300 ppm CO₂) has a mass fraction of water in the bubble of 0.71. If I don't allow for CO₂ mobility, I find that the pressurization and heat increase give me $\epsilon_g = 0.03$ and $\Delta\epsilon_g = -0.02$ quite close to what you report in Fig. 2A. However, if I solve for the joint H₂O-CO₂ mass conservation, I find $\epsilon_g = 0.044$ and $\Delta\epsilon_g = -0.006$, much less resorption than you calculate. In this case, the solubility only increases to 4.70 wt% H₂O and 419 ppm CO₂ with a mass fraction of water in the vapor of 50% versus the water-only mass balance equation which gives 4.83 wt% H₂O and 51% water in the vapor. It seemed from the methods description that the numerical simulations are solving the coupled H₂O-CO₂ problem, so you need to do the same for the analysis of driving mechanisms.

Lines 110-113 & 120-132: I found the text here a little unclear about the difference between mechanisms 2 & 3. My understanding is that you mean for 2) to be about the change in solubility due to melt chemistry from a combination of mixing and crystal melting, but you also discuss the reduction in total water on a per volume basis by introducing the drier magma (and also from crystal melting?), while 3) is about the decrease in vesicularity to accommodate the volume expansion of crystal melting and crystal-poor recharge (which are reduced by allowing the chamber to expand as your simulations do). But, it took me a few reads through the paragraph and figure 2 and the supplement to understand this so maybe make the distinction a little more apparent. And which of these exactly do you mean by "dilution"?

I also don't understand how you calculated mechanism 2. Is this the heating effect on solubility? I would expect heating to matter more than your values suggest and would cause exsolution, not resorption. Is this the change in solubility with changing chemistry (you could lose a couple wt% silica), but then how are you calculating that given that the Liu et al. (2005) solubility law doesn't account for the melt chemistry. Using my calculation from earlier, but now only heating without pressurization I exsolve either $\Delta\epsilon_g = 0.005$ for the mixed H₂O-CO₂ or $\Delta\epsilon_g = 0.017$ for the immobile CO₂. Also, in Fig. 2B, they all have a change of $\Delta\epsilon_g \approx -0.029$, with little variability in there but not passing through 0 gas change when the solubility change is 0. So where is that coming from if you're isolating only the component from solubility?

Regime diagram

It's a bit odd here to have a plot for the scaling with really only one dimensionless parameter and then plot things along the 1-to-1 line or not. From just mass balance:

$Min = \partial/\partial t(\rho V)$ and defining compressibility, $\beta = 1/\rho \partial\rho/\partial p$, such that $\rho = \rho_0 \exp(\Delta p)$ you can find:

$Min = \rho(\beta \partial p/\partial t V + \partial V/\partial t)$, which suggests the scaling $1 \sim \rho\beta V / (Min \partial p/\partial t) + \rho/Min \partial V/\partial t$. You plot the first term divided between the axes, and suggest that the deviation from the 1-to-1 line is the result of "amplified pressurization" which "can result from reduced bulk magma compressibility as the MVP volume decreases during a return to volatile-undersaturated conditions." Compressibility is in your term so it should be explicitly accounted for (although you have not specified how it was calculated). Instead, it looks to me like you are seeing the influence of the second term which is the change in volume of the chamber with time which should scale in some way with the visco-elastic response of the crust. That is, when you lose the vapor phase, the compressibility of the magma chamber becomes comparable to the compressibility of the host rock (or maybe this is accommodated by deformation at the surface through the crust rheology), you could find the relative contribution of these from the model governing equations.

Compatibility with Aso-4 observations

The motivating petrological study from the first author (<https://doi.org/10.1016/j.epsl.2023.118400>) suggests a zoned magma chamber and volatile undersaturation only in the late-erupted products. Additionally, you require very fast recharge rates to push the magma chamber to undersaturated conditions. Taken together, these suggest to me that undersaturation may be only a local and probably transient effect during pulsed mafic recharge; the claim of undersaturated conditions reducing the compressibility of the bulk magma chamber allowing for accelerated pressurization and earlier eruption seems unlikely to me. The fully undersaturated model results require the fastest recharge rates and an initial condition of the magma chamber at 4 wt% H₂O which I would argue is in fact incompatible with your observations of 4.4-4.7 wt% H₂O, which should be better matched by the 4.5 wt% H₂O simulations that don't produce undersaturation. In your magnetite paper (<https://doi.org/10.1016/j.jvolgeores.2023.107789>) you also calculate the temperatures of the pre-Aso-4 eruptions to be ≈ 925 °C and early Aso-4 erupted material to be ≈ 870 °C, so I'm having trouble reconciling your conceptual model of a cool, large reservoir being heated by an increase in the recharge rate over the 5 kyr gap between eruptions. I don't think there's a big inconsistency between the data and a zoned reservoir with resorption and recharge taking place in the lower zone (late-

erupted material) while the upper zone is cooling and evolving over the repose period, but I would then expect that upper portion to have a co-existing vapor phase and remain compressible. I understand the numerical model assumes a mixed reservoir and so you have to push it really far to get the resorption and undersaturation that maybe would be easy to produce in a zoned chamber. You could argue that it might exist in another system, but the examples you give in Lines 173-174 also show observations that span the water solubility calculation as the Aso-4 samples seem to.

What are the noteworthy results?

The authors demonstrate several mechanisms by which mafic recharge could produce volatile resorption, which I agree is potentially an overlooked process. However, the model requires quite high recharge rates (accounting for about 10% of the total volume) and initial conditions very close to saturation (an open system?) before it can produce undersaturated conditions in the entire magma chamber. While I agree that an undersaturated magma chamber would have low compressibility and could change the eruptability, I think this case is probably not so widespread and not apparent in the selected case study.

Will the work be of significance to the field and related fields? How does it compare to the established literature? If the work is not original, please provide relevant references.

This work could point the community to have a better discussion about the saturation state of magmatic reservoirs and how that saturation might vary in space and time. The authors could point more directly in the discussion to the implications this would have for petrologists studying these systems and what they should look at (maybe for length at the expense of the hazard implications which I think are not so strong).

Does the work support the conclusions and claims, or is additional evidence needed?

The analysis of the mechanisms should be improved and while the model can produce the behavior the authors describe, I think it is not as widespread as they claim and doesn't match their case study.

Are there any flaws in the data analysis, interpretation and conclusions? - Do these prohibit publication or require revision? Errors in the calculations for Figure 2 (see above), which are only for the discussion and are simple to correct, but may change the conclusions. I expect the calculation is correct in the numerical simulations that provide the data for the other figures, although in that case the authors may not have identified the most important processes.

Is the methodology sound? Does the work meet the expected standards in your field?

The numerical simulations are already established by this authorship team and collaborators. The model is simple enough to enable running many simulations, but has the appropriate complexity to answer the question posed in this text.

Is there enough detail provided in the methods for the work to be reproduced?

There is some detail missing. I wasn't able to determine how the calculations for Fig. 2 and Fig. 4 were done. While I understand the temperature for each model was set according to the desired crystallinity and water content and so varies, it would be helpful to report the range of initial temperatures explained in the text and add the temperature for each run to the relevant supplementary data tables. Same for the bulk composition (even just SiO₂) for the recharge material.

Minor points:

Lines 99-101: "We find that while higher recharge temperatures and lower recharge volatile contents promote resorption, recharge rate is the main control on this process." I think you should be clear here that the scaling dependence should be (I think) approximately linear for all of these factors, you're just varying recharge rate over a much large range, which is reasonable, but just be clear about that in the wording here. Unless you really think the pressurization is the driving mechanism.

Line 192-194: "we expect a resorption-induced reduction of the MVP to cause a decrease in magma compressibility, which could enhance of fluctuate ground deformation prior to large silicic eruptions." I don't think this claim is that simple. True, it would decrease the compressibility, but the resorption process itself actually decreases the magma volume so there's a competing effect here. You're right in the case of complete resorption, but anything shy of that you would need to actually look into which factor wins.

Method line 416: This is a very limited depth range. I understand you are interested in the very large systems, but for the small magma chambers, it would be interesting to see the effect of a broader range. You don't really present the effect of depth, but I could imagine that changes in the solubility sensitivity to pressure at different depths could be interesting.

Methods line 417: Please give the range of temperatures investigated.

Method line 419: The listed CO₂ contents don't match the table.

Method line 423: You don't give the composition for the recharge melt. Is it supposed to be a basalt? You give a melt density for the rhyodacite which is the most evolved end-member.

Supplementary information 1 line 41: 0.09 is too high for your simulations, and also not what you present in Fig. 2C. Maybe 0.009? Also, not sure why these calculations were done on the basis of 10% initial crystallinity if the simulations are for 15%.

Sincerely,
Janine Birnbaum

(Remarks on code availability)

I did not have access to the code repository.

Version 1:

Reviewer comments:

Reviewer #1

(Remarks to the Author)

I have read the revised version of that paper and found that the authors have correctly addressed the comments raised during the first round of review.

I would simply recommend the authors to change widespread process by common process in their abstract (I would even write ..likely a common process). I don't think we have the data to say that this paper would represent a paradigm shift in our perception of the mechanical evolution of magmatic reservoirs.

The excess sulfur conundrum documented for several explosive silicic eruptions is generally explained by the existence of a separate fluid phase, which, if true, shows that in those cases at least, volatile resorption did not play a crucial mechanical role.

I understand that to get published in journals of wide audience one has to show that the findings provided are of wide/global relevance but here it is mostly the attention to a under-appreciated mechanism which makes the paper appealing and interesting. In my opinion it is now the job of future studies to test that possibility before we can claim it is a general one.

Other than that I have no more comments to do and therefore recommend acceptance of this paper which addresses a mechanism which was so far not considered in the modelling reservoir evolution.

Bruno Scaillet

(Remarks on code availability)

Reviewer #2

(Remarks to the Author)

I thank the authors for their engagement with my comments and questions. I appreciate the detail of their responses in the rebuttal and I am satisfied that they have made sufficient changes to the manuscript text and figures to address them. I am now happy to recommend this manuscript for publication, and hope that it sparks fruitful further discussions and new avenues of research by its readers.

(Remarks on code availability)

Reviewer #3

(Remarks to the Author)

NCOMMS-25-51869A

Volatile resorption expedites eruption onset in large silicic systems

Keller, F., Townsend, M., Troch, J., Huber, C.

Main comments

Resorption drivers calculations:

1) Has this calculation now been removed from the methods?

2) The discussion of these is still a bit confused – going back and forth in order between the dilution of magma mixing and crystal melting

3) I think the new discussion is a cleaner in terms of the simplicity of the calculations, which is not per se a problem. But, I think there are two overlooked (competing) effects that need to be directly addressed in this discussion: you mix the water contents in process 2, but not the CO₂ contents which should have the opposite effect. It's fine to keep them very simple, treat them separately and show the effect of keeping water constant and essentially just adding some CO₂ (CO₂ flushing?) It seems from your supplementary data that the more CO₂ you add, the less resorption you get. Additionally, you do just the pressure effect, please also do just the temperature effect.

4) I asked on the first draft and I'm still confused. Why in your analysis do you only account for adding 1% of chamber volume through recharge when your undersaturated simulations add 10%? Eruptions occur for 10-2.4 km³/yr * 5x10³ yr = 20 km³ which is 4% of the total chamber volume, and you explore up to 10-2 km³/yr * 5x10³ yr = 50 km³ (10% of 500 km³). This will bring the effect closer to what you get for 5% crystal melting (albeit that melt is completely anhydrous rather than 1 wt% water). Or is it that they erupt before getting there?

Figure 3:

From this plotting it is not apparent to the reader what difference is from the resorption/exsolution and what is from the recharge rate. Can you do this in normalized time (time*recharge rate) which would just be added volume so that we can distinguish?

Additionally, something looks odd to me in the calculation of average compressibility. Why is the average compressibility for run 3 between the value for runs 1 and 2 when it has uniformly higher vesicularity and lower pressure?

I'm skeptical of this extrapolated curve. How was this fit? Run 2 also shows a certain acceleration despite remaining volatile saturated. If you adopt the scaled axis you could compare two runs that go from just volatile saturated to just undersaturated for a more direct comparison.

Compressibility calculation:

Since it is a critical parameter, how is the compressibility being calculated? It seems to me that from a volume-averaged-type

formulation, the increase in compressibility at low volume fraction of exsolved volatiles into undersaturation should have a smooth, rather than stepped, increase in compressibility. Why would this produce a sharp change in the pressurization rate? Also, the difference in compressibility is so small between the exsolving and volatile undersaturated runs (4×10^{-11} 1/Pa) which for 20 MPa of pressure should only be a volume difference of 0.08%. It seems to me that it's really just the volume reduction of the gas phase itself that's buffering the pressure, more than the "compressibility". I understand that in a bulk sense these are indistinguishable, but the discussion should be clear that this isn't about the instantaneous suspension compressibility if that's the case.

Scaling argument (Fig. 4):

I still think it would be better to have a non-dimensional factor given that the pressure term should be combined with the rest. But interesting that including the volume change didn't help. That suggests to me that you haven't actually identified the driving process since compressibility alone can't explain your result!

See attached document for line-by-line comments.

[Editorial Note: See end of file for attachment]

Best regards,
Janine Birnbaum

(Remarks on code availability)

Given that the model is well-established, I didn't run the code, but I can see the files and inputs are available at the listed repository and all the data for the figures is provided in the supplementary material.

Version 2:

Reviewer comments:

Reviewer #1

(Remarks to the Author)

I have had now time to read carefully the last version of the paper, as well as the accompanying rebuttal letter.

It is my opinion that the authors have done a good job in clearly and carefully addressing/considering the majority of the comments raised during the several rounds of reviews.

There are obviously still open questions but in my opinion they pertain to the scientific debate. Those questions may even promote future work on the subject, in particular additional modelling efforts, which will deal with more complete/complex treatment of the physical process addressed here (for instance what would be the role of the geometry of reservoir, or its depth, which may affect the tensile strength of host rocks).

I therefore recommend acceptance.

Bruno Scaillet

(Remarks on code availability)

REVIEWER COMMENTS

Reviewer #1 (Remarks to the Author):

This is an interesting paper that explores the role of pressurisation of a magma reservoir upon recharge. Using an established code (ie an already published one by the same group) the authors explore how the excess fluid/gas that a reservoir may hold prior to recharge may be resorbed in response to the pressure increase. The authors show that under some conditions of magma recharge, that excess fluid may be fully redissolved in the melt, which results in a significant decrease of its bulk compressibility, hence in an increase of the rate of pressure increase, which in turn may bring the reservoir to its failure at a faster rate as well. It is true that the conventional wisdom in this area is to consider that cooling increases excess fluid amount hence magma eruptability. This study shows that this may not always be the case and on that ground it provides an interesting and novel view on this complex problem. This study is a logical follow up of previous efforts led by the same team (Keller et al., 2013 in EPSL) in which the authors proposed the idea of a shift between gas saturated and under-saturated conditions of the Aso reservoir (using apatite volatiles systematic), and suggested that this could affect eruption frequency. In the present contribution they carry out the numerical simulations testing their hypothesis, which they find is a physically tenable one.

Being not a numerical modeller I can only trust what the model predicts, but the core code has been already used in a number of published papers, so I assume it has been already benchmarked and it is working properly.

My comments are mostly suggestions, clarifications asked for, and some are possibly naive questions given my limited experience in numerical simulations:

Response:

We thank reviewer 1 for their constructive and insightful feedback, we really appreciate it and believe it helped us to improve the manuscript!

Below we provide detailed answers to the reviewer's comments:

1. The authors investigated a range of volatile contents, either for the resident or the intruded magma. For the later, it is my understanding that it aims at simulating conditions relevant to a mafic recharge (as inferred for Aso by previous work, in particular Kaneko et al., 2007). My question is why the bulk CO₂ of that mafic end-member is restricted to 1000 ppm or less? There is evidence that the CO₂ content of mafic magmas, including in arcs, can reach several wt% (see the case of Stromboli for instance). It would be interesting to expand the CO₂ range to at least 1 wt% bulk content (as in Scholz et al., 2023) and see how this could affect model's outcome.

Response:

We initially limited our simulations to a recharge CO₂ content of 1000 ppm, as CO₂ plays a less important role in controlling bulk magma compressibility than H₂O (see Scholz et al., 2023 - <https://doi.org/10.1029/2023GC011151>) and because we don't necessarily regard the recharge as strictly basaltic but more as intermediate to basaltic-andesitic, where CO₂ contents are expected to be lower overall.

To fully address the reviewers' concerns, we ran an additional suite of simulations including 1 wt% CO₂ content in the recharging magmas. The results are depicted in the updated figure 1F and also in supplementary fig. 1D. We see that higher recharge CO₂ does not impact whether resorption occurs but rather influence its efficiency – i.e. at the same change in crystallinity, less volatile resorption takes place in runs with higher recharge CO₂. As such high CO₂ contents increase the mass of the MVP, we don't observe resorption to fully undersaturated conditions in these runs.

2. I also guess or understand that the role of recharge is to provide heat and mass (including H₂O and CO₂?) to the reservoir but only heat is allowed to propagate upwards (in addition to that lost to the host rock)? In other words the melt fraction trends which are shown in extended data Figure 2 are those which are used to simulate the bulk of the magma reservoir? Or do you take into account also the mafic layer? Similarly, are the H₂O and CO₂ sourced from the mafic recharge transferred into the overlying silicic melt? I would also suggest the authors to compare their curves with experimental data to assess how accurate is MELTs in calculating melt fraction at a given bulk H₂O and CO₂ content: for instance the phase equilibria of Costa et al (2004) have been performed on a whole rock composition very similar to that selected in the present work (see also Scaillet and Evans 1999, though this uses a slightly more mafic magma).

Response:

This question might address two points: 1) the general model setup and how mixing takes place within this setup, and 2) crystal-melt curves used to parametrize the model.

- 1) *Generally, our model does not implement the composition of recharge. As a 1D box model, the model simulates recharge by continuously adding mass, heat and volatiles into the silicic magma chamber and assumes immediate and complete mixing between the host and recharge magmas. Therefore, we cannot model individual silicic and mafic layers unfortunately. What we can do is control the volatile content and the temperature of recharge that is being added into the silicic reservoir, where we choose water and CO₂ contents typical for intermediate to mafic magmas. As these magmas are entirely mixed with the host magma, volatiles are indeed being transferred. We find though that the volatile content and temperature of the recharging magmas is secondary to the recharge rate – so lower recharging volatile contents can ‘dry out’ the magma more efficiently, i.e. more volatile resorption at similar crystallinity changes, but resorption only occurs under high recharge rates (see Figure 1C, Supplementary Fig. 1).*
- 2) *We added the data from Costa 2004 to supplementary fig. 2 and find that MELTS generally predicts lower temperatures during crystallization, but the slopes of the curves are very similar to the ones produced in the Costa experiments, which is more essential to the model to predict crystallinity changes.*

3. From what I understand, the compressibility of host rocks β is kept constant in all simulations. Presumably this property is likely to vary between volcanic areas, depending on the level of intrusion, how fractured are the host rocks etc..The authors fix compressibility to $\beta=1/10-10$ Pa (correct?). Quoting Gudmundsson (Volcanotectonics, 2020, page 286) it is said that host rock compressibility can be $\beta=3 \times 10^{-11}$ Pa⁻¹. I have no opinion on which value is the most appropriate but a comment of why the value 1/10-10 Pa has been selected would be welcome. I am asking this because using my own rudimentary code which simulates the evolution of internal pressure of a cooling magma reservoir, made me understand that the host rock bulk modulus is quite important in determining how much of a pressure increase arises (and not only the magma compressibility).

Response:

Indeed, host rock bulk moduli can vary between 10^{10} Pa to 10^{11} Pa - we use a value of 10^{10} Pa here. This value is dependent on crustal composition with some sedimentary rocks (e.g. sandstone) sometimes being as low as 5×10^9 Pa (see Jaeger, Cook, Zimmermann, 2007, page 190), while more mafic rocks (e.g. islandic crust) are often closer to 10^{11} Pa (see your reference from Gudmundsson, 2020). In addition, this value is also impacted by the presence of fractures in the crust, where fractures usually lower the bulk modulus even of more mafic rocks to roughly 10^{10} Pa.

4. One of the great appeal of the model, for a non expert like me, is that it allows to simulate the time evolution of the reservoir, which is a highly desirable property for any model aimed at anticipating eruptions with some confidence. However, the age of Aso-4 event seems to come from Hbl Ar dating (Albert et al., 2019 quoted by Keller et al., 2023), but that phase is described as being xenocrystic or not at equilibrium with its host melt (Ishibashi et al., 2018) which is supported by the lack of Hbl in the run products of

hydrothermal experiments of Ushioda et al (2020). I wonder therefore how sure one can be about the time interval between Aso-Y and Aso-4? Is there any other data or way to ascertain the age of Aso-4?

Response:

We are not sure if we agree that amphiboles are in disequilibrium with the Aso-4 melts. In our 2021 study, we evaluated the mineral assemblage of the Aso-4 eruption and found pretty striking evidence that the amphibole should be in equilibrium with the host Aso-4 melts – e.g. the amphiboles are euhedral and do not show any visible disequilibrium structures or resorption rims, they have relatively high REE contents suggesting crystallization from evolved REE-enriched melts and their Eu-anomaly is progressively increasing as expected during co-crystallization with plagioclase. Therefore, we suggest that the hbl age is reliable.

Nevertheless, several alternative age constraints were provided before the K-Ar age from Albert et al., 2019. For example, Aoki 2008 used sediment cores from the Pacific to estimate the Aso-4 age between 86.8 and 87.3 ka, while a recent study by Hoshizumi et al., 2022 (in Japanese) used loess chronometry to estimate the Aso-4 age to 88 ka (the Aso-Y age also stems from the Hoshizumi study). So overall all these ages fall in similar ranges, making us confident that the time interval between Aso-Y and Aso-4 is reliable.

5. One important result or at least, petrologically speaking, the one that brings question to my mind is that the reservoir may undergo a significant pressure increase during cooling (as in previous work). Is this pressure increase documented (aside from apatite volatile record)? I mean, in an ideal world, one could expect to see phenocrysts hosting melt inclusions showing such a pressure increase, ie an increase of both CO₂ and H₂O contents as MI get closer to the rim? I know this is difficult task (in particular in phenocryst-poor rocks and given the rather modest changes in pressure) but in my opinion it would be the real smoking gun of the process advocated here. Also, if I understood correctly, volatile resorption comes mostly from an increase in pressure but not from a decrease in crystal fraction? It would help the reader to show a graph with the evolution of these two parameters (gas and crystal fractions) with time during a single but representative simulation that culminates in gas-free conditions close to reservoir rupture. Increasing pressure while cooling and preserving no textural evidence of melting/crystallisation requires quite a subtle combination of parameters. According to petrological constraints (Keller et al., 2023, JVGR), the temperature drop between Aso-A and Aso-4 is about 55°C. Does that fit with the model output?

Response:

Absolutely agree, it would be incredible if such changes could be traced in melt inclusions! We don't have such data available at this point, but we also hope to stir new data collection with this publication. So hopefully such work could be part of new research avenues. Another question is also on what timescales resorption occurs and if enough time for MI formation is given. Such questions could potentially be approached using diffusion chronometry in the MI. However, these questions unfortunately go beyond the focus of this study for now.

We added an additional figure (Supplementary figure 3) showing the progression of T, P, ϵ_g , ϵ_v , V, and a plot of the change in T vs. the time until eruption. What we can see is 1) pressure changes substantially (~20 MPa) while temperature might change only a couple of degrees, and 2) resorption can occur over various timescales up until almost 2500 yrs. However, resorption until full undersaturation usually occurs very fast, mainly controlled by the fact that pretty small volumes of gas are present at simulation start.

We further looked more into comparing temperature changes to our Aso case study. Unfortunately, we do not have reliable T constraints from the Aso-Y eruption, so we cannot fully reconstruct the temperature increase between Aso-Y and Aso-4. As you mentioned before between Aso-A (97.7 ka) and Aso-4 a temperature drop of 55°C occurred. However, we assume that the actual drop happened already between Aso-A and Aso-Y, which we do not model here. However, we know that the mineral assemblage of Aso-Y contained significant amounts of biotite, which according to our MELTs simulations starts crystallizing at ~870°C at 200 MPa and H₂O of 5 wt%. So, the temperature increase can be subtle between Aso-Y and Aso-4 and in our view is realistic. In addition, we find sieved plagioclase cores and reverse zoning patterns in plagioclase and amphibole in the late-erupted material of the Aso-4 event (see Keller et al., 2021), which implies that some reheating likely took place.

6. The model considers the EOS derived by Halbach and Chatterjee on H₂O to simulate the PVT of the gas phase. This EOS is for H₂O only and it seems that it is used to compute the volume properties of the H₂O-CO₂ fluid mixture. Am I correct? If so, any idea of how a more real EOS (fluid mixture) would affect the results? More specifically, I don't understand how equilibrium distribution of both H₂O and CO₂ species between gas and melt can be computed with an EOS for water only?

Response:

If we understand this comment correctly, it might be pointing to two individual processes within the code. The EOS from Halbach and Chatterjee is mainly used to calculate the density of the gas phase and how it changes with pressure (i.e. the compressibility) and temperature (i.e. the thermal expansion). In exemplary run batches (n=1800) >95% of the runs with an exsolved MVP have XCO₂ <<0.1 and therefore the gas phase is heavily water-dominated and the error propagated through using an EOS including only H₂O is small enough to neglect.

The partitioning of H₂O and CO₂ between the MVP and melt is then calculated using the H₂O-CO₂ solubility law developed by Liu et al., 2005. Here both components are accounted for, which we agree is necessary to accurately determine the behavior of H₂O and CO₂ in the melt and gas phase.

7. I think that Bishop Tuff may not be a good example of a system sitting near gas saturation. According to Wallace et al (1995,1999) seminal work, this system records good evidence of being already gas saturated. As for Campi Flegrei, I don't believe we have the firm evidence in hand for such a shift toward gas undersaturation, since temperature estimates of that system are not that well constrained in my opinion.

Response:

Thank you for pointing out the Wallace references, these are really interesting. They might actually support our hypothesis. If we get it correctly, the main message from the Wallace papers is that the Bishop Tuff had gradients in exsolved gas from early-erupted material that contained higher quantities of H₂O dominated gas to late-erupted material, which contained significantly less gas, which at the same time was enriched in CO₂ (even though still H₂O dominated). This is actually pretty similar to what we see in the natural Aso-4 samples, where undersaturation is observed in the late-erupted material. Wallace et al explain the accumulation of gas in the early-erupted material of the Bishop Tuff through H₂O enrichment during silicic magma differentiation and density stratification, while CO₂ enrichment in deeper parts is suggested to result from CO₂-rich recharge. We think that resorption could have similar signatures – we assume that the effects of recharge (e.g. pressurization, heating, dilution) are locally stronger in deeper parts of the reservoir making resorption more efficient in these parts. Such a process could contribute to the stratification of the Bishop Tuff. In addition, CO₂ enrichment in the Bishop Tuff would lead to an increase in the XCO₂ of the MVP, this is something we observe in our runs.

The main point we wanted to make with these examples was that a large number of silicic caldera systems are sitting around saturation (not excluding that they already exsolved an MVP but still remain close to saturation so that the MVP volume is small), which is also supported by the H₂O contents reported in Wallace (5.3-5.7 wt%), which are very close to saturation that we roughly calculated to be around 5.6 wt% using the Liu et al., 2005 solubility model.

As for Campi Flegrei, we used data provided in Forni et al., 2018 to argue for a recharge-driven increase in storage temperatures. These are equilibrium temperatures calculated using the cpx-liquid thermometer from Masotta et al., 2013.

8. Pre-eruptive declines in gas emissions also arise from progressive sealing of the volcanic conduit, and not only because of the high viscosity of the magma

Response:

That's a good point; we added this to the manuscript.

9. I don't think it is correct to say that CO₂ solubility is higher in mafic melts relative to more silicic ones (line 186). CO₂ solubilities are not that much different in mafic and felsic melts, and in fact, a rhyolite

could dissolve even more CO₂ if held at the same fCO₂ and T than a basalt liquid. What happens in nature is that by the time silicic melts are produced most of the CO₂ of the system has been lost.

Response:

Thank you for catching this. We corrected our statement here saying that we expect SO₂ to be more soluble in the silicic melt than CO₂, therefore leading to an increase in the CO₂/SO₂ ratio with beginning resorption.

10. From what you have done, I think that the best pre-eruptive signal of a pressurising system entering a gas-free regime would be a break in slope of geodetic signals recording ground uplift (following what is shown in your figure 3b(inset)), assuming the time series of data is long enough to capture this change? Yet, the Campi Flegrei example shows that the rapid up and down ground motions are not that easy to interpret in terms of deep seated processes.

Response:

We agree that interpreting the signals is very hard. We mainly tried to suggest possible monitoring signals, in which MVP resorption might be captured, but more research certainly needs to be done on that. By including our statements here, we primarily want to encourage readers to consider such research.

Bruno Scaillet.

Reviewer #1 (Remarks on code availability):

The code has been used several times by the same team, and has led to several publications which I assume have permitted to check it

Reviewer #2 (Remarks to the Author):

Review of “Volatile resorption expedites eruption onset in large silicic systems” by Keller, Townsend, Troch, and Huber.

I have read this manuscript closely and with great interest. The authors use the case study of the caldera-forming Aso-4 eruption to investigate the hypothesis that mafic recharge of large silicic magma chambers can cause resorption of exsolved magmatic volatiles, and that this resorption can then cause increased chamber pressurization rates through the corresponding decrease in magma compressibility, resulting in earlier eruption onset. The authors use thermal mechanical modelling in conjunction with previously published geochemical data regarding the pre-eruptive chemistry, volatile contents and magma chamber conditions of the Aso-4 eruption. Their results indicate that when recharge rates are high volatile resorption can occur, primarily through chamber pressurization (which increases volatile solubility in the melt) and increase of the melt volume fraction by addition of hot recharge melts that also act to melt existing crystals in the resident magma (which together adds relatively volatile-poor melt thus reducing the average volatile content of the melt, approaching a state of volatile undersaturation). The authors suggest that elevated pressurization rates caused by complete volatile resorption could have brought forward the onset of the Aso-4 eruption by ~70 years. The authors propose that detecting precursory signals of chamber pressurization caused by volatile resorption could be critical for early warning of future eruptions.

Overall I am convinced that volatile resorption may be a common (or, not uncommon) process in large silicic magma chambers and that a consequence of the resorption process can be increased rates of chamber pressurization that may trigger eruption. I agree it is an important topic and worthy of future study. However, I am left uncertain about the extent of the authors’ argument regarding volatile resorption as an eruption trigger, and unconvinced by how useful or meaningful it is that the Aso-4 eruption may have been ‘expedited’ by up to 100 years. This latter point is a significant issue for a paper with the current manuscript title.

The paper is generally well-written and logically set out, however I was still left with many questions about the authors' precise meaning. I have included these in my comments below and I expect that most of them can be addressed by tighter and more explicit text, particularly for figure captions, and perhaps some minor adjustments to the figures. Addressing them will avoid unnecessary confusion for the reader, particularly for the manuscript layout that provides few figures in the main text with detailed methods at the end. The most important revisions will be those addressing to what extent and why the authors are arguing for the importance of volatile resorption in understanding large silicic eruptions. If these can be addressed convincingly I would recommend the manuscript for publication in Nature Communications.

Response:

We thank reviewer 2 for their thoughtful and supportive review. We agree that the manuscript was very strongly focused on the early eruption onset hypothesis, and we tried to address this issue by reframing the manuscript slightly to point out that we do not assume that resorption is an eruption trigger itself. We rather see it as a process that occurs simultaneously with high recharge rates (acting as a trigger) and might be important to achieve critical overpressures and destabilization of large silicic magma chambers that are usually very difficult to pressurize.

We provide more details on these changes in our detailed replies below and hope that the manuscript is more clear now!

I have listed my detailed comments below. Line numbers refer to the line numbers of the PDF version provided for review.

Title

'expedites eruption onset'. Is this the most important finding? The slightly earlier eruption?

Response:

*Overall, we think that uncovering MVP resorption as a possible and not even uncommon process in silicic systems is the main message for us. So far, recharge paired with volatile **exsolution** has been interpreted to play a crucial role in triggering silicic eruptions. By introducing volatile resorption (occurring coupled to high recharge rates), we argue that the opposite process to volatile exsolution might actually play a significant and so far, overlooked role in chamber destabilization of often fairly stable silicic reservoirs.*

We do think the expedited eruption onset is a key finding in our study. When comparing model results for resorbing cases to cases that exsolve volatiles, pressurization rates are generally significantly higher, even if the magma does not fully return to undersaturation (see for example figure 3). This is mainly of course due to the elevated recharge rates, however, resorption also decreases the compressibility additionally accelerating pressurization. We think that such expedited pressurization rates play a crucial role in reaching the critical overpressure for eruption onset, yet we believe that this is more a supporting process to allow silicic systems to pressurize critically than an actual eruption trigger.

Abstract

L16 What is the meaning of 'independent' here? Previously published?

Response:

Yes, we refer to previously published; as we updated the abstract, this section is not included anymore.

L16 and throughout: kyrs is a length of time. An eruption 86 thousand years ago should be 86 ka or 86 kyrs BP, etc

Response:

Thank you for pointing this out, we corrected this throughout the manuscript!

Main text

L36-56 this is a nice (clear, concise) explanation of the study context

Response:

Thank you, we are happy to hear that we could capture the context understandably!

L52 are vapor loss by passive degassing/viscous relaxation of the crust included in the magma chamber model used? The model description includes ‘in a visco-elastic crust’ (L76), what about passive degassing?

Response:

No, passive degassing is not considered here. We also don't think that passive degassing played a major role in controlling the MVP loss observed in the Aso reservoir. Even if passive degassing takes place, the MVP fraction will not go back to zero (Parmigiani et al., 2017 – <https://doi.org/10.1002/2017GC006912>, or Degruyter et al., 2019 – <https://doi.org/10.1098/rsta.2018.0017>), therefore, apatite would have continued to capture a water saturated signature from the melt rather than showing an undersaturated signature, as we find for Aso-4.

L55-56 “The high compressibility of the MVP increases bulk magma compressibility, dampening pressurization from recharge and leading to less frequent, yet larger, eruptions”. Reading this, I wonder if the proposed volatile resorption mechanism should have any expected effect on 1) frequency and 2) size of eruptions? Certainly Aso-4 was caldera-forming even though undersaturated... would it have been even bigger if the initial magma was not undersaturated?

Response:

This is a very good point, thank you. Given that Aso-4 was a caldera forming eruption, we don't think that resorption significantly impacted the erupted volume in this case, as it is mainly driven by the collapse of the reservoir roof. We believe that resorption might rather play an important role in destabilizing the reservoir through enhanced pressurization so that large-scale silicic systems can be brought out of equilibrium and achieve eruption. As these systems are usually thermally mature, reaching the critical overpressure for eruption is significantly more difficult than in smaller silicic systems and it is not yet very well understood what can trigger large-scale caldera eruptions (hypotheses range from roof collapse to volatile exsolution, see the introduction). We believe that resorption induced by increasing recharge rates prior to eruption might be a missing puzzle piece.

However, we do think that in smaller silicic systems resorption might indeed cause smaller volume eruptions (see Townsend et al., 2020 - <https://doi.org/10.1130/G47045.1>). Regarding frequency - as resorption goes along with high recharge rates, the eruption frequency is generally high in systems that undergo resorption, however, we do believe that reactivation of a reservoir is easier following an eruption that underwent resorption as reservoir cooling during eruption is limited, therefore it remains less viscous and can be pressurized for eruption more readily again.

L61 another example where kyrs should be ka or kyrs BP etc

Response:

Thanks, we corrected throughout the manuscript.

L64 water undersaturated. Was CO₂ saturated i.e. CO₂ bubbles present in the initial magma? Or the argument is the eruption could start without any MVP at all?

Response:

We argue that the eruption could have been triggered without an MVP. We mainly refer to water-saturation here as apatite is predominantly capturing H₂O partitioning, while our model incorporates a mixed H₂O-CO₂ volatile phase. However, we also argue that water is the more common and compressible volatile phase and therefore has a more important effect on overall magma compressibility and magma dynamics

(see Scholz et al., 2023). Rhyolites are usually considered relatively CO₂ poor due to early-stage degassing of CO₂ at deeper levels of the crust.

L64 why is passive degassing of the MVP ruled out? If the answer is in the other Keller et al paper please add it here concisely for the reader

Response:

Passive degassing is mainly ruled out because apatite records water undersaturation from the magma. This would not be the case if passive degassing was the primary driver for volatile reduction because, as mentioned above, passive degassing rates are generally too slow to diffuse out the entire MVP volume fraction (see references above). At the same time, any accompanying crystallization of a mostly anhydrous mineral assemblage would further drive up water contents in the melt, and keep the magma saturated. OH groups would continue to partition into the (rare) bubbles and OH partitioning between apatite and the melt would not be impacted, continuing to capture water-saturated signatures in apatite. Therefore we conclude that passive degassing alone cannot account for the transition to MVP-undersaturation and an additional process must be at play here.

We added a little clarification here in the manuscript.

L66 kyrs is used correctly here

Response:

Thanks!

Volatile dynamics prior to the Aso-4 caldera-forming eruption

See L52 comment above, does model include passive degassing effects

Response:

See our comments above – the model does not include passive degassing; however, we also think that it did not play a major role in controlling volatile loss due to signatures captured in apatite.

L81-82 recharge magma is ‘drier and hotter’. Is it mafic? Is the recharge melt composition included in the magma mixing modelling? And is it accurate to call it ‘drier’ or is ‘less volatile-rich’ better – what about CO₂ as well as H₂O?

Response:

That’s a good point; recharge at Aso is mostly basaltic andesite in composition. We cannot prescribe a fixed composition to the recharge in the model, we rather determine temperatures and volatile content to mimic more mafic recharge (i.e. hotter and drier recharge usually). We renamed ‘drier’ to ‘less volatile-rich’ recharge in the main text following your suggestion.

Figure 1 caption is confusing. Do all three panels show data with the identical initial parameters as in the first sentence of the caption? Are the data in B and C for the same runs, i.e. this figure shows the results of varying resident magma H₂O wt% and recharge rate and nothing else? It took me some detailed re-reading to reach this (hopefully correct) understanding. Panels B and C easily look to a quick glance like they are plotting temporal variations in crystallinity & MVP during three runs, with start point in upper right and end point in lower left. Are the symbols showing the final change in crystallinity & MVP at the end of each run? Why are some ‘undersaturated’ diamonds shown within Regime 2 – are these runs where the MVP totally resorbed but then some volatiles exsolved again before the model run ended? In general it is better to have a lengthy caption to prevent unnecessary confusion for readers seeing this diagram for the first time. I am not a fan of ‘show don’t tell’ in figure captions – better to ‘show and tell, then let the reader decide if they agree’.

Response:

Sorry that the figure caption seemed confusing – in trying to keep everything as compact as possible we seem to have lost important information on the way. Yes, your interpretation is absolutely correct, both

panels B and C show the exact same data just color-coding changes for water content (B) and recharge rate (C). We remodeled the figure following your next comment and tried to make the figure caption clearer. As for the diamonds in regime 2 ('resorption and cooling') – we expect to see some very few points in this regime, where resorption itself is predominantly driven by pressurization of the reservoir but without adding enough material to really heat up the whole magma chamber.

Related, Extended Data Figure 1 was extremely helpful in understanding this manuscript and the meaning of Fig. 1 and it is unfortunate it is not in the main manuscript. Could it not be combined to make the original Fig. 1 a 6 panel figure?

Response:

Thank you for the suggestion, we added a modified version of the extended data fig. 1 here in panels D-F. The main difference between the new panels and the extended data fig. 1 is that here data is filtered to represent only the cases that contain 4 wt% initial H₂O and 100 ppm CO₂ in the resident magma, while the recharge rate is varied through 100 logarithmically spaced steps between 10⁻⁴ to 10⁻² km³/yr and not kept constant at the endmember values.

L93 “we define four distinct regimes that describe the coupled thermal and volatile saturation conditions in the chamber as a function of recharge rate (Fig. 1A)”. There is nothing in Panel A to indicate the influence of recharge rate on these regimes, and the recharge rate is argued to be a crucial factor in the volatile resorption mechanism. Can useful schematic arrow(s) of ‘increasing recharge rate’ be applied to this regime illustration? Or again, it is an argument to combine the Extended Data Figure 1 here.

Response:

This is another great suggestion. As mentioned above, we added extra panels to the figure. To not overload panel A, we refrained from adding additional arrows here but added arrows for increasing recharge rate in panels C and D for clarity.

L98 Regime 4 is not observed for the Aso data, why not? What conditions would need to be met? Is it possible at other volcanic systems or is it an unnatural scenario?

Response:

We have not observed any data falling into regime 4 in all our simulations. Generally, this regime ('Exsolution and Heating') might also be difficult to achieve as MVP exsolution is naturally driven by crystallization and cooling. We could imagine that replenishment of very volatile-rich and hot recharge at high rates might drive an increase in the exsolved MVP fraction and simultaneously heating the reservoir, but at the same time a rapid depressurization might be required, which does not seem to be very likely to occur in nature.

L108 most(?) of the model runs that resulted in eruption within the 5 kyr model timeframe were not fully undersaturated (i.e. not diamond symbols). Why is it so important that Aso-4 was back to undersaturated conditions? What difference (eruption size, duration?) between the runs that were undersaturated vs partially resorbed but still with MVP? Is the authors' argument that magma chambers that enter Regime 3 will have eruptions sooner than those that stay in Regimes 1-2 (or 4?), even if the MVP is not fully resorbed? Or is full MVP resorption considered to cause the most rapid return to eruption?

Response:

Thank you for pointing this out – yes cases that return to full undersaturation would trigger eruptions faster than those that keep some of the MVP, i.e. undersaturation accelerates pressurization additionally. This is also visible in Figure 4, where all cases that return to undersaturation are indeed recording elevated pressurization rates. But overall resorption still triggers eruptions faster than exsolution due to decreasing bulk magma compressibility associated with resorption.

The main point why it is important for Aso to have returned to undersaturation – at least in parts of the reservoir - is to match the observations we made with apatite data recording undersaturated conditions.

We are basically aiming to understand the magma dynamics using the model here that led to the observations made in Aso.

What drives volatile resorption during magma recharge?

Processes 2 and 3 need clarification, between the description in the text here and the caption of Figure 2.

Response:

We added a substantial amount of information to the main text, hopefully the process becomes a bit clearer now.

What are the 5 data points shown in Figure 2? Is the change in MVP vol% by these difference processes being output from model runs like those in Figure 1, or just a simple calculation of ‘suppose Process X causes an overall change in parameter Y of amount Z’?

Response:

Figure 2 is independent of the model runs. The points represent isolated calculations to determine the amount each process (pressurization, mixing, and changes in phase proportions) can contribute to resorption. A detailed explanation is given in the supplementary information, and additional information is added to the manuscript as pointed out above.

Process 2:

Line 111 “(2) mixing of silicic resident magma with drier and hotter recharge magma”, but also Line 128 “The effect of chemical mixing between the resident and recharge magmas (2)”, and also Fig 2B caption “(B) magma mixing of silicic resident magma with drier recharge magmas (Δ Solubility in wt%)” (Line 461).

What is being included in the calculation for Process 2, as shown in Fig 2B as ‘delta solubility wt%’?

Change in solubility value caused by change in SiO₂ content of resident magma by mixing of initial rhyodacite with mafic recharge? (Does this study’s model include the chemistry of the magma? If not, why not – if it is shown to be negligible, say so).

And/or the change in solubility value caused by temperature increase due to addition of hotter recharge magma?

As for ‘drier’, solubility value of a volatile in a given melt composition at a given P,T condition is fixed. Mixing one dry (volatile-poor) version of a melt into a wet (volatile-rich) version of that melt does not change the solubility value. So it seems ‘drier’ should be deleted from the description of Process 2. Instead of ‘delta solubility’, are the authors aiming for a meaning of ‘closer to/further from saturated conditions’? This would more accurately reflect the Process 2 mixing described in Line 129-130 (unless, there is an unspoken assumption about different temperature and composition of the recharge vs resident magmas).

Response:

Thank you for pointing out some of the issues with process 2. We redid the calculations using a modified version of the equation considering changes in water mass and mass of the magma in the chamber. So, we now specifically test how mixing of a less volatile-enriched magma with a volatile-enriched host magma impacts the water mass in the chamber and the chamber volume throughout recharge. Therefore, we do not present delta_solubility anymore but show how the gas fraction changes with volume in the magma chamber as more recharge is added. We hope this is clearer now.

The model does not directly account for changes in the SiO₂ content of the magma, but it does account for changes in X_{CO2}. We followed your advice above and renamed the recharge from ‘drier’ to less volatile-rich.

Process 3:

Line 112 “(3) variations in phase proportions of crystals, melt and MVP through influx of less crystalline recharge magma”

Fig 2C caption “(C) changes in crystal volume fraction ($\Delta\epsilon_x$) induced by the dilution of resident magma with anhydrous recharge and crystal-derived melts.” (Line 462)

It seems heat (hotter recharge magma) is missing from these outlines of Process 3, although the effect of heat for melting crystals and liberating anhydrous melt is indeed described in the following text.

Response:

We here assume that recharge does both, add crystal-poor recharge magma itself, which increases the melt fraction, i.e. dilutes the host magma, and adds anhydrous melt through crystal melting. However, we only capture an increase in overall melt fraction as it would be difficult to account for the individual processes.

Why doesn't Fig 2B have change in gas fraction going from 0 on the x axis, like panels A and C? Why does 'reduced solubility' (as indicated by the y axis) still correspond to decreases in gas fraction – lower solubility should mean more of the volatiles in the system should be exsolved in bubbles?

Response:

As mentioned above, we redid the calculations for process 2. With this setup now, the change in gas fraction is 0 if there is no change in solubility and volume.

The implications of volatile resorption on eruption timing

Fig. 3 explain the notation in the panels in the figure caption.

Response:

We clarified the annotations in the figure caption.

Line 468 'inset' typo

Response:

Thank you for pointing this out, we corrected the typo.

The three eruptions in Fig 3 all happen on much shorter timescales than the 5 kyrs between Aso-Y and Aso-4 that is(?) being modelled. Are these scenarios realistic?

Response:

That's a great question. Yes, we are modelling the time frame between the Aso-Y and Aso-4 eruption here, and yes, we believe that these are realistic scenarios, even though the times over which resorption occurs is significantly shorter than the 5 kyrs time frame. What this mainly tells us is that a change in recharge rate must have occurred at some point prior to the Aso-4 event, which induced resorption and allowed for more efficient pressurization of the reservoir, ultimately inducing eruption. As we cannot really tell when such a change in recharge rate occurred, this model gives us an estimate on potential time frames. We tried to make this clearer in the manuscript.

Line 147-148 The finding that Aso-4 eruption occurs 70 years earlier when the MVP is totally resorbed rather than partially resorbed (but still in Regime 3) seems to be the big crux of this paper. But with timescales of thousands of years since the last eruption, does this plus/minus 70 years have any practical value? We are not able to make eruption predictions at anything like that level of precision (if at all).

Response:

Indeed the 70 yrs earlier eruption does not seem to be a lot of time. However, on a human timescale such times are actually considerable, and it would be great to be able to forecast that a system might be prone for eruption within the next few decades. We think of resorption as an indicator that a system is highly recharged and that pressurization rates are climaxing, making it more prone for eruption. Therefore, resorption might be a warning sign for elevated pressurization rates, calling for enhanced monitoring.

In general, the bigger point we are trying to make is that resorption is a viable process that can occur in nature, which so far has not been considered to the best of our knowledge and is the opposite process of volatile exsolution which is often cited together with recharge as an eruption trigger.

Line 162-164 “We find that once a system transitions from volatile-saturated to undersaturated conditions, a drop in magma bulk compressibility amplifies pressurization expediting chamber destabilization and earlier eruption”.

Is the change to undersaturated conditions (no MVP) the critical point for ‘expediting’ eruptions? As said above, an earlier onset of 70 years does not seem such a remarkable finding. What about Regime 3 in general? If a large silicic magma chamber is experiencing high recharge rates, does it experience an earlier eruption if its initial conditions (density, size etc) cause it to evolve through the Regime 3 pathway (even if the MVP is not totally resorbed) compared to if its initial conditions led it to evolve through Regimes 1,2 or 4? This might be of greater impact in understanding which systems could erupt quickest and necessitate closest monitoring of temporal variation of precursory signals.

Response:

Thank you for pointing this out. Definitely, the onset of resorption in general does already expedite pressurization of the system and with that eruption onset. Therefore, whenever a system goes through regime 3, an eruption might occur faster than in regime 1. We tried to make this clearer in the manuscript text.

Line 167 again I am not sure to what extent the authors are arguing for the importance of resorption as an eruption trigger. Does this trigger require total resorption of the MVP or is only partial resorption enough? Or is their argument that these big systems (Bishop Tuff, KPT, CF) require volatile undersaturation to erupt; their high recharge rates are not enough to trigger eruption unless they also trigger undersaturation?

Response:

We don't necessarily argue that resorption is an eruption trigger. Our idea is more that high recharge rates that are often implied for large-scale silicic systems might not go along with the volatile exsolution process that is often being suggested as eruption trigger. We find that exsolution is not as efficient in triggering eruptions and that rather resorption occurs as a result of high recharge rates. The resorption induced changes in compressibility then accelerate pressurization of the system and might explain how thermally buffered systems reach their critical overpressure.

L187 what is the reasoning for the expected increase in CO₂/SO₂ ratio? If CO₂ becomes more soluble as the system becomes more mafic (through mixing of mafic recharge), then more CO₂ dissolves and CO₂/SO₂ ratio should instead decrease. Or CO₂ solubility is higher in mafic melts thus the mafic recharge is bringing in lots more CO₂ to the chamber hence higher CO₂/SO₂ – but how is this different to the case of mafic recharge that doesn't trigger resorption?

Response:

Thank you for pointing this out, our reasoning of a CO₂/SO₂ increase was mixed up here. We believe that once resorption occurs SO₂ goes back into the melt, while CO₂ remains in the MVP, therefore increasing the CO₂/SO₂ ratio. This is actually a process that we also see in the XCO₂, which increases as a result of resorption during our model runs. In the end, this process is not really different from adding CO₂ from a more mafic recharge magma, which could complicate the interpretation of increasing CO₂/SO₂ ratios at active volcanoes. We think however, that isotopic ratios of the MVP might be affected differently by resorption, yet, as we are not really specialists in gas monitoring of volcanoes it is hard for us to make predictions. It would be really interesting though to see future work on that!

L192-193 The link between the volatile resorption mechanism and the observable precursory ground deformation and perhaps seismicity signals that might result seems like a more useful and significant outcome for hazard mitigation than the finding that resorption might cause eruptions to occur a few decades

earlier than they otherwise would. It needs a lot more work, probably beyond the scope of this study, to know if this is feasible.

Response:

We agree that such work would be beyond the scope of our manuscript. We primarily mention these outlooks here, as we would love to see more research in this direction in the future from specialists in the respective fields. We believe that such research would be important as resorption might actually give us significant warning signs of upcoming eruptions, so understanding its effects on monitoring signals and tracing its occurrence would be really interesting.

L194 the reference for the CF deformation is for recent observations. Are the authors implying these deformations could be happening because the CF magma chamber is currently undersaturated? Do the CF CO₂/SO₂ or other observations over the same timescale support their interpretation?

Response:

That's an interesting thought. We believe that resorption is a result of strong recharge events, so we would expect ground deformation to be mainly driven by input of large quantities of magma, which could be intensified by volatile resorption. Assuming that an existing MVP buffers uplift, the loss of the MVP would have the contrary effect therefore supporting strong uplift. For CF that would mean that we expect a sharp uplift event that is associated with a pulse of new magma input as observed in the 1982-84 uplift (~1.86 m in less than 3 yrs, which accounts for a recharge rate on the order of 10⁻² km³/yr). Changes in gas composition are a little more difficult to predict. As mentioned above, it will be hard to distinguish resorption from input of CO₂-rich fluids, as we expect predominantly water to diffuse back into the melt, leaving the MVP CO₂ enriched (we see an increase in the mole fraction of CO₂ accompanying resorption). Following Chiodini et al., 2012 (<https://doi.org/10.1130/G33251.1>) the 1982-84 recharge was preceded by a decrease in the CO₂/H₂O ratio indicating the influx of new magmatic fluids – during the following sharp uplift period though, the CO₂/H₂O ratio significantly increased. The authors interpreted this increase as contribution from CO₂-rich fluids from depth; however, we think that this sharp increase could also be a sign of ongoing resorption. We added a short sentence about these observations in the manuscript, however, since we are no experts in understanding gas emission data, we prefer to keep these observations on the suggestive side.

L197-198 “identify highly recharged systems that undergo amplified pressurization with the potential of near-future eruptions”

Again I am left wondering what this means. Systems with the potential for total MVP resorption? Or partial MVP as well? And the meaning of ‘near-future’. The three eruption scenarios in Fig 3 all take place within ~300 to 2500 years. Is this realistic? And what type and size of eruption will result? See also my comment on L55-56.

Response:

Thank you for pointing out these open questions! We believe that we can use resorption as an indicator for systems that undergo high recharge rates. We find that if resorption occurs pressurization rates are generally higher than if the system is undergoing exsolution, therefore we think that already the onset of resorption in general, without hitting undersaturated conditions, is already a sign that the system is undergoing enhanced pressurization. If a transition to undersaturation is actually met, it is very likely that the system will probably experience eruption soon, given that recharge will be continued at the established high rates.

Concerning the time frame, we believe that the time intervals given for resorption in fig. 3 are realistic. As it is usually difficult to assess the risk posed by especially large-scale silicic systems, we think that even such time frames are helpful, in concentrating monitoring efforts. With over >100 active silicic caldera

only in arc settings, knowing which ones have the higher potential for renewed eruption, will be already helpful in our opinion.

Methods

L359 do different run conditions have different erupted magma volumes? Is volatile undersaturation exclusively associated with large caldera-forming eruptions?

Response:

Yes, we do find that different run conditions produce different erupted volumes, this is mainly the effect of time – the longer the system has to accumulate volume, the more mass is erupted during the event. When looking at the new Supplementary Fig. 3, we show 2 different runs, the first one going back to undersaturation has a significantly smaller erupted mass (5.5×10^{12} kg) than the second run (3.1×10^{13} kg), which experiences eruption only after 2500 yrs. Therefore, we don't think that undersaturation is directly linked to caldera formation, we rather believe that resorption is an overlooked process that could explain how we destabilize caldera-forming systems that are otherwise really difficult to pressurize to the point of failure.

L363 Aso-specific compositions. The following calibration for the Aso-Y crystallinity with T, P, H₂O, does not include the effect of mixing in mafic magma with low SiO₂. Is this change in overall melt chemistry shown to be insignificant for the melting of crystals etc that promotes resorption?

Response:

Exactly, the calibration only uses the Aso-Y specific composition and does not take lowering of SiO₂ contents through mixing into account. In other words, the recharging magma has the same assumed composition as the resident magma, but through setting temperature and volatile conditions of the recharging magma to variable conditions, we can mimic more mafic conditions (e.g. hotter, H₂O poorer and CO₂ richer than the host magma). Though lowering the SiO₂ content of the magma can impact the equilibrium conditions of crystals already present in the chamber, we believe that temperature is the primary driver for crystal melting in our runs. In addition, a portion of changes in melt fraction also stems from the addition of crystal-poor recharge magma – if the temperature of the recharging magma is higher than the host, the crystallinity of the recharging magma will be adjusted to these higher temperatures and therefore will be less crystal-rich.

L423-424 if recharge is mafic it doesn't make sense to use the density of Aso-Y rhyodacite melt

Response:

This is a valid point, we used the same density here, as we assume that the composition of the recharging magma is the same as the resident magma for simplicity of the model. The more mafic conditions are then simulated through higher temperatures and variable volatile contents compared to the resident magma. We clarified this in the main text by referring to less volatile-rich magma rather than more mafic magma.

Reviewer #3 (Remarks to the Author):

Summary

The authors build upon previous observational results of mineral chemistry from the Aso-Y and Aso-4 eruptions to consider the implications of mafic recharge in large silicic systems on volatile saturation state and the consequences on eruptability. Simulations using an existing numerical model solve for the evolution of the state of the magma chamber, including pressure, temperature, and chemistry and how perturbations to these change the volume fractions of crystals and vapor which have an impact on the magma chamber compressibility. While I think the core argument that mafic recharge could induce some degree of volatile resorption is sound, I have questions about the details and the strength of the authors' claims of magma chamber-wide undersaturation before the manuscript is ready to be published.

Response:

We thank reviewer 3 for their detailed review. We provide answers to each point brought up by the reviewer below:

Distinguishing mechanisms of resorption

Overall, I like the approach taken to tease apart the multiple possible mechanisms to identify the most important. However, I think there are some issues in the calculation:

1) the assertion for constant chamber volume ($V_0=V_i$) is, I think, causing a problem here. The simulations with volatile resorption presented in Fig. 1 clearly show a change in chamber volume. For example, the high flux and 4.5 wt% initial water case; $\Delta\epsilon_g \approx -0.03$ and $\Delta\epsilon_x \approx -0.035$, by mass conservation of the melt $\Delta\epsilon_x \approx \Delta\epsilon_m$, $\rho_x=2800$, $\rho_m=2250$ and $\rho_g=350$ then $(-0.03*350 - 0.035*2800 + 0.035*2250)/(0.05*350+0.15*2800+0.8*2250)$ then the chamber volume is changing by ~1% which could be important here given the small initial bubble volume. Also, you can see with the joint constraint of $\Delta\epsilon_g + \Delta\epsilon_m + \Delta\epsilon_x = 1$ (which is true by definition) that you are forcing the decrease in the gas volume fraction (and therefore absolute gas volume) to occur at the expense of crystal and melt volume, which is probably why you come to the conclusion that the crystal melting (and resultant volume expansion of the crystal+melt assemblage) is the important driving factor.

Response:

Thank you for looking into our approach so carefully. We agree that the reviewer's point are generally really good and valid, however, we want to emphasize that we use the process calculations to estimate hierarchy of the three processes and their contribution to resorption, rather than accurately reproducing the integrated changes in gas fraction produced in the model. We therefore anticipate closed system conditions (i.e. $V_0=V_i$ and $Mw_0 = Mwi$) to calculate the isolated effect of changes in pressure and phase proportions.

Following your suggestions, we adjusted our calculations of process 2 (mixing) by allowing for open system conditions, so we are now accounting for changes in the water mass and volume of the system. We agree that in the mixing case this will play an important role, as recharge is adding water mass and volume to the system, which we previously overlooked. Using this approach, we indeed find that magma mixing can resorb up to 8% of the MVP, therefore pressurization and changes in phase proportions, remain the main process driving volatile resorption. We updated our calculations in the supplementary files and updated figure 2. We additionally caught a sign error in the equations, therefore the numbers presented in fig. 2 are slightly different than in the previous version.

2) you're solving here for only the water mass balance between the bubbles and melt, but in the mixed H2O-CO2 system, the CO2 really matters. If you have 4.5 wt% H2O and 300 ppm CO2 total, the vapor would have 30 vol% CO2 (50 wt%). Which has a direct effect on the vapor volume and you need to consider the resorption of CO2 to get the right vapor volume. But also, CO2 has a strong effect on the water solubility. I ran a quick mass balance calculation (ignoring the input magma for hypothesis 1) for joint water and CO2 starting from:

$$M_{H2O} = \rho_m(1-\epsilon_g, 0-\epsilon_x, 0)_{meq, H2O} V_0 + \rho_{-H2O} \chi_{-H2O} \epsilon_g V$$

$$M_{CO2} = \rho_m(1-\epsilon_g, 0-\epsilon_x, 0)_{meq, CO2} V_0 + \rho_{-CO2} (1-\chi_{-H2O}) \epsilon_g V$$

Where χ_{-H2O} is the mass fraction of water in the bubble. And from a little rearrangement, assuming no change in the melt and crystal volumes: $V = [1 - \epsilon_g, 0 + \epsilon_g(1 - \epsilon_g, 0)/(1 - \epsilon_g)]V_0$. I don't have your exact numbers, but if I choose a pressurization from 200 MPa to 220 MPa along with a temperature increase from 875 °C to 913 °C (for addition of melt at 1300 °C at a rate of 10-2 km3/yr, although the result is similar in the iso-thermal case) and 15% crystallinity along with $\rho_{-H2O,0}=381$ kg/m3, $\rho_{-H2O,i}=389$ kg/m3, $\rho_{-CO2,0}=922$ kg/m3, $\rho_{-CO2,i}=982$ kg/m3 using the ideal gas law for CO2 and the Pitzer and Sterner (1994) equation of state for water and the Liu et al. (2005) mixed solubility model, I find that a starting

vesicularity of 0.05 and starting melt composition of 4.55 wt% H₂O and 370 ppm CO₂ (equilibrium for 4.5 wt% H₂O total and 300 ppm CO₂) has a mass fraction of water in the bubble of 0.71. If I don't allow for CO₂ mobility, I find that the pressurization and heat increase give me $\epsilon_g=0.03$ and $\Delta\epsilon_g=-0.02$ quite close to what you report in Fig. 2A. However, if I solve for the joint H₂O-CO₂ mass conservation, I find $\epsilon_g=0.044$ and $\Delta\epsilon_g=-0.006$, much less resorption than you calculate. In this case, the solubility only increases to 4.70 wt% H₂O and 419 ppm CO₂ with a mass fraction of water in the vapor of 50% versus the water-only mass balance equation which gives 4.83 wt% H₂O and 51% water in the vapor. It seemed from the methods description that the numerical simulations are solving the coupled H₂O-CO₂ problem, so you need to do the same for the analysis of driving mechanisms.

Response:

Thank you again for putting the effort in to go through our approach in detail, we really appreciate the effort. We agree that CO₂ is playing an important role, however, most of our run scenarios have XCO₂ values well below 0.1 leaving the MVP strongly water dominated. As we are trying to approach the hierarchy of mechanisms driving resorption, we believe that focusing on water, which is the dominant volatile species and has a larger impact on magma compressibility (see Scholz et al., 2023), is sufficient for these approximations – again we are not attempting to reproduce integrated changes of gas fraction produced by the model.

As mentioned above, we redid our calculations of process 2 now accounting for changes in volume and mass.

Lines 110-113 & 120-132: I found the text here a little unclear about the difference between mechanisms 2 & 3. My understanding is that you mean for 2) to be about the change in solubility due to melt chemistry from a combination of mixing and crystal melting, but you also discuss the reduction in total water on a per volume basis by introducing the drier magma (and also from crystal melting?), while 3) is about the decrease in vesicularity to accommodate the volume expansion of crystal melting and crystal-poor recharge (which are reduced by allowing the chamber to expand as your simulations do). But, it took me a few reads through the paragraph and figure 2 and the supplement to understand this so maybe make the distinction a little more apparent. And which of these exactly do you mean by “dilution”?

Response:

We clarified the processes in the manuscript. Process 2 calculates how magma mixing of a drier recharge with a wetter host magma impacts resorption, while process 3 accounts for increasing the melt fraction of the host magma through addition of crystal-poor recharge melt and crystal melting.

I also don't understand how you calculated mechanism 2. Is this the heating effect on solubility? I would expect heating to matter more than your values suggest and would cause exsolution, not resorption. Is this the change in solubility with changing chemistry (you could lose a couple wt% silica), but then how are you calculating that given that the Liu et al. (2005) solubility law doesn't account for the melt chemistry. Using my calculation from earlier, but now only heating without pressurization I exsolve either $\Delta\epsilon_g=0.005$ for the mixed H₂O-CO₂ or $\Delta\epsilon_g=0.017$ for the immobile CO₂. Also, in Fig. 2B, they all have a change of $\Delta\epsilon_g\approx-0.029$, with little variability in there but not passing through 0 gas change when the solubility change is 0. So where is that coming from if you're isolating only the component from solubility?

Response:

We redid the calculations for processes 2 as described above accounting for variations in the magma volume and water mass rather than solubility. We hope this clears the reviewers concerns.

Regime diagram

It's a bit odd here to have a plot for the scaling with really only one dimensionless parameter and then plot things along the 1-to-1 line or not. From just mass balance:

$\text{Min}=\partial/\partial t(\rho V)$ and defining compressibility, $\beta=1/\rho \partial\rho/\partial p$, such that $\rho=\rho_0\exp(\beta \Delta p)$ you can find:

$\text{Min}=\rho(\beta \partial p/\partial t V + \partial V/\partial t)$, which suggests the scaling $1\sim\rho\beta V/(\text{Min } \partial p/\partial t) + \rho/\text{Min } \partial V/\partial t$. You plot the first term divided between the axes, and suggest that the deviation from the 1-to-1 line is the result of “amplified pressurization” which “can result from reduced bulk magma compressibility as the MVP volume decreases during a return to volatile-undersaturated conditions.” Compressibility is in your term so it should be explicitly accounted for (although you have not specified how it was calculated). Instead, it looks to me like you are seeing the influence of the second term which is the change in volume of the chamber with time which should scale in some way with the visco-elastic response of the crust. That is, when you lose the vapor phase, the compressibility of the magma chamber becomes comparable to the compressibility of the host rock (or maybe this is accommodated by deformation at the surface through the crust rheology), you could find the relative contribution of these from the model governing equations.

Response:

The scaling law we used here is a common scaling law used in our field (see e.g. Townsend et al., 2019 - <https://doi.org/10.1029/2018GC008103>). We followed the reviewers' recommendations and tried to implement a second term, however, this term does not help to bring the data together along the 1-to-1 line, but rather enhances the differences. We expect that increasing volume rather shifts the points to the left in the diagram space, therefore producing the opposite results. We also already previously checked whether crustal viscosity would have an impact on the data distribution, but we could not find any correlation.

We calculated the compressibility following the approach of Townsend & Huber 2020 (<https://doi.org/10.1130/G47045.1>) for the effective compressibility.

Compatibility with Aso-4 observations

The motivating petrological study from the first author (<https://doi.org/10.1016/j.epsl.2023.118400>) suggests a zoned magma chamber and volatile undersaturation only in the late-erupted products. Additionally, you require very fast recharge rates to push the magma chamber to undersaturated conditions. Taken together, these suggest to me that undersaturation may be only a local and probably transient effect during pulsed mafic recharge; the claim of undersaturated conditions reducing the compressibility of the bulk magma chamber allowing for accelerated pressurization and earlier eruption seems unlikely to me. The fully undersaturated model results require the fastest recharge rates and an initial condition of the magma chamber at 4 wt% H₂O which I would argue is in fact incompatible with your observations of 4.4-4.7 wt% H₂O, which should be better matched by the 4.5 wt% H₂O simulations that don't produce undersaturation. In your magnetite paper (<https://doi.org/10.1016/j.jvolgeores.2023.107789>) you also calculate the temperatures of the pre-Aso-4 eruptions to be ≈ 925 °C and early Aso-4 erupted material to be ≈ 870 °C, so I'm having trouble reconciling your conceptual model of a cool, large reservoir being heated by an increase in the recharge rate over the 5 kyr gap between eruptions. I don't think there's a big inconsistency between the data and a zoned reservoir with resorption and recharge taking place in the lower zone (late-erupted material) while the upper zone is cooling and evolving over the repose period, but I would then expect that upper portion to have a co-existing vapor phase and remain compressible. I understand the numerical model assumes a mixed reservoir and so you have to push it really far to get the resorption and undersaturation that maybe would be easy to produce in a zoned chamber. You could argue that it might exist in another system, but the examples you give in Lines 173-174 also show observations that span the water solubility calculation as the Aso-4 samples seem to.

Response:

Thank you for pointing out these concerns. We find that the onset of resorption is very much dependent on the initial crystallinity of the resident magma. The observed crystallinities in the whole ignimbrite range from as low as <5% to up to 50% in the most crystalline samples. We chose a crystallinity of 15% to represent a good average of the chamber, given that volumetrically most of the ignimbrite represents crystal-poor material. However, if we choose lower crystallinities undersaturation is achieved in magmas

that have 4.5 wt% H₂O. Given the uncertainties of the model and the uncertainties stemming from hygrometers themselves (here +/- 0.35 wt%), we in fact argue that the water contents are matching.

For our magnetite paper, we are referring to Aso-A as the last eruption before Aso-4 – this is because Aso-Y was first discovered and described in the Hoshizumi 2022 study, therefore we did not have access to Aso-Y samples during this study. We included Aso-Y for the first time in the apatite study of 2023. Therefore, the temperature variations you are referring to are between the Aso-A eruption before Aso-Y and the Aso-4 event. Unfortunately, we do not have reliable T constraints from the Aso-Y eruption, so we cannot reliably check the temperature increase between Aso-Y and Aso-4. However, we know that the mineral assemblage of Aso-Y contained significant amounts of biotite, which according to our MELTs simulations starts crystallizing at ~870°C at 200 MPa and H₂O of 5 wt%, implying almost no change in temperature between Aso-Y and Aso-4. We also want to emphasize that the late-erupted part of the Aso-4 ignimbrite contains sieved plagioclase cores and reverse zoning patterns in plagioclase and amphibole (see Keller et al., 2021). This implied that some reheating must have taken place.

The recharge rates we are using here are referenced in literature – see Schöpa and Annen 2013 (<https://doi.org/10.1002/jgrb.50127>) or Costa 2008 ([https://doi.org/10.1016/S1871-644X\(07\)00001-0](https://doi.org/10.1016/S1871-644X(07)00001-0)) for examples. Based on such studies, it is widely accepted that such high recharge rates (even exceeding 10⁻² km³/yr) are required to grow caldera-forming reservoirs and their eruption. We therefore argue that our inputs here are reasonable.

We agree that modelling the whole magma chamber is not ideal for representing a zoned chamber, however, we also believe that our results are applicable. As we described in the manuscript, the driving factor of resorption is pressurization induced by recharging of the magma chamber. Pressure increases will be felt throughout a zoned reservoir, as mass is added to the system increasing the internal pressure. Therefore, resorption could occur throughout the reservoir. We agree that locally, it might be enhanced by additional effects such as heating and changes in phase proportions, and these local changes might be the factor pushing the system back to undersaturated conditions. If recharge continues under such conditions critical overpressurization can be reached more easily, helping to trigger such large-scale eruptions, that otherwise remain difficult to destabilize. Overall, we want to make clear that yes, resorption is a process that is initiated by high recharge fluxes, but we believe that the implications of resorption, i.e. lowering the compressibility of the magma – even locally- are important to drive pressurization of the system and reach the critical overpressure faster than in exsolving systems.

What are the noteworthy results?

The authors demonstrate several mechanisms by which mafic recharge could produce volatile resorption, which I agree is potentially an overlooked process. However, the model requires quite high recharge rates (accounting for about 10% of the total volume) and initial conditions very close to saturation (an open system?) before it can produce undersaturated conditions in the entire magma chamber. While I agree that an undersaturated magma chamber would have low compressibility and could change the eruptability, I think this case is probably not so widespread and not apparent in the selected case study.

Response:

Thank you for sharing these concerns. We appreciate hearing that the reviewer agrees that resorption might be an overlooked processes. As we explained in our comment above, we do think that the recharge rates used here are reasonable, as they are based on frequently cited literature and recharge rates of up to 10⁻² km³/yr are high, yet not unrealistic values (looking at the 1982-84 unrest period at Campi Flegrei – we have an estimated addition of 0.025 km³ material accounting for a recharge rate of ~10⁻² km³/yr – see Troise et al., 2019). In addition, such rates given in the literature are typically long-term averages integrated over long timescales, yet realistically recharge rates vary over time, so very likely there will be shorter episodes of high recharge rates exceeding these average values, and other periods with lower recharge rates than the averages. Especially in the case of Aso, high recharge rates could have occurred on a short timescale, making the chosen recharge rates very realistic.

We are not quite sure on what evidence the 10% volume change resulting from such recharge referred to by the reviewer is based on. Referring to the new supplementary fig. 3, we see a volume increase of 1% in our longest run, which we believe is a realistic scenario.

It is correct that the conditions to achieve undersaturation have to be very close to saturation, however, we want to emphasize that resorption itself, without achieving undersaturation, already induces accelerated pressurization of the system. This is of course tied to the high recharge rates but also the fact, that compressibility starts to decrease.

Our model results suggest that resorption almost always occurs under high recharge rates (in the 15vol% initial crystallinity case as soon as recharge rates exceed 10^{-3} km³/yr). Assuming that large caldera systems are expected to experience enhanced recharge rates (see references mentioned above and in the manuscript) and considering that recharge rates might be pulsed rather than long-time continuous input, we think that such episodes of high flux pulses are common and that the process can be in fact widespread. Of course, this is a hypothesis, and we would love to see more research on this topic to get more evidence-based input from more than one case scenario. This is one of the main purposes of this paper, to get attention to this potential process and test the frequency of its occurrence in nature.

Will the work be of significance to the field and related fields? How does it compare to the established literature? If the work is not original, please provide relevant references.

This work could point the community to have a better discussion about the saturation state of magmatic reservoirs and how that saturation might vary in space and time. The authors could point more directly in the discussion to the implications this would have for petrologists studying these systems and what they should look at (maybe for length at the expense of the hazard implications which I think are not so strong).

Response:

We agree, we would love to spark more discussion about saturation levels in magmatic systems prior to eruption. We already give instructions on how petrologists might look for this process in nature and how systems of interest could be identified in the conclusions and implications section.

Does the work support the conclusions and claims, or is additional evidence needed? The analysis of the mechanisms should be improved and while the model can produce the behavior the authors describe, I think it is not as widespread as they claim and doesn't match their case study.

Response:

We understand the reviewers concern here, however, in our previous comments we lined out why we think that the reviewers concerns might be misled. The water contents cited here are well within the range of error of the water contents determined in Aso samples (taking model and hygrometer uncertainties into account) and the temperature variations addressed by the reviewer previously referred to the Aso-A instead of the Aso-Y eruption, which is not the focus of this study. Therefore, we do believe that our case study is matching.

We further agree that very specific conditions are required to achieve a system transition to undersaturation, however, we do mention a number of systems that have water contents close to saturation, which therefore might have been subject to similar processes as Aso. We also would like to bring the study of Brookfield et al., 2023 to the reviewer's attention – here the authors suggest a similar sweet spot condition around solubility to achieve high overpressures in magma chambers. In the end, large-scale silicic eruptions are very rare events, and specific conditions might be required to achieve such rare natural events. Finally, we also want to emphasize again that we believe that resorption is a process that is generally important, even if undersaturation is not reached – referring to figure 3, we see that resorption is generally more efficient in pressurizing a system compared to exsolution, which is of course also tied to higher recharge rates. As such high recharge rates are commonly suggested for large-scale silicic system, resorption might also be a common process. As we deliver the groundwork on detecting resorption in natural data and can reproduce it with our model, we want to spark an interest for more research on this topic in the future, more petrological data will deliver the evidence for whether resorption is widespread or not.

Are there any flaws in the data analysis, interpretation and conclusions? - Do these prohibit publication or require revision?

Errors in the calculations for Figure 2 (see above), which are only for the discussion and are simple to correct, but may change the conclusions. I expect the calculation is correct in the numerical simulations that provide the data for the other figures, although in that case the authors may not have identified the most important processes.

Response:

Thank you for pointing out these concerns, we addressed the calculations in detail above and approached the calculation of especially process 2 – magma mixing from a new perspective. With these changes and identifying a previous sign error in our equation, we still find that pressurization followed by changes in the phase proportions are the main contributors to volatile resorption.

Is the methodology sound? Does the work meet the expected standards in your field?

The numerical simulations are already established by this authorship team and collaborators. The model is simple enough to enable running many simulations, but has the appropriate complexity to answer the question posed in this text.

Response:

Thank you.

Is there enough detail provided in the methods for the work to be reproduced?

There is some detail missing. I wasn't able to determine how the calculations for Fig. 2 and Fig. 4 were done. While I understand the temperature for each model was set according to the desired crystallinity and water content and so varies, it would be helpful to report the range of initial temperatures explained in the text and add the temperature for each run to the relevant supplementary data tables. Same for the bulk composition (even just SiO₂) for the recharge material.

Response:

We supplied additional information on the calculations for figure 2 in the supplementary materials, we hope it is clearer now.

Regarding initial temperatures – initial temperatures for the 15 vol% case usually vary between ~855 to ~900 °C. We added this in parentheses in the parameter space section of the methods. We, however, would like to refrain from adding all initial temperatures to the supplementary files, initial temperatures can be easily assessed with the code available on Zenodo. In addition, Supplementary fig. 3 was added giving examples of temperature curves during resorption.

Minor points:

Lines 99-101: “We find that while higher recharge temperatures and lower recharge volatile contents promote resorption, recharge rate is the main control on this process.” I think you should be clear here that the scaling dependence should be (I think) approximately linear for all of these factors, you're just varying recharge rate over a much large range, which is reasonable, but just be clear about that in the wording here. Unless you really think the pressurization is the driving mechanism.

Response:

We do believe that pressurization induced by high recharge rates is indeed the driving factor of resorption – this is also what is shown with our calculations of figure 2. In addition, we varied the recharge volatile contents and temperatures over ranges that are comparable to values found in natural systems, therefore the variations we receive should also be on the order of magnitude of the expected outcomes. For us this means that indeed, the recharge rate has more important controls than the other factors – which yes, is in parts due to the naturally larger variety of recharge rates than the other parameters.

Line 192-194: “we expect a resorption-induced reduction of the MVP to cause a decrease in magma compressibility, which could enhance of fluctuate ground deformation prior to large silicic eruptions.” I

don't think this claim is that simple. True, it would decrease the compressibility, but the resorption process itself actually decreases the magma volume so there's a competing effect here. You're right in the case of complete resorption, but anything shy of that you would need to actually look into which factor wins.

Response:

We agree that this statement is a simplification, we added a sentence here in the manuscript to point out the competing effects of volume decrease and lowered magma compressibility. However, we want to make clear that in this section, we are brainstorming potential effects on how resorption could be tracked in nature rather than providing definite statements. Our main goal is to spark new research, which could give the answers to our hypothesis mentioned here, therefore, we would like to keep the statement.

Method line 416: This is a very limited depth range. I understand you are interested in the very large systems, but for the small magma chambers, it would be interesting to see the effect of a broader range. You don't really present the effect of depth, but I could imagine that changes in the solubility sensitivity to pressure at different depths could be interesting.

Response:

Absolutely, we agree that a wider depths range would be interesting especially for smaller scale systems. However here we oriented our study on Aso to which these depth ranges refer and generally magma chambers for large-scale silicic systems are expected to be at a depth around 200 MPa (see the study of Huber et al., 2019 <https://doi.org/10.1038/s41561-019-0415-6>). Future research on the depth effect would be really interesting to see but is beyond the scope of this study.

Out of interest, we also ran a quick simulation (using a 10^{-2} km³/yr recharge rate, 4 wt% initial H₂O, 100 ppm CO₂) showing that high recharge rates would have similar effects on shallower magma chambers. It mainly seems like resorption is more efficient in shallower and smaller chambers, which is probably related to higher proportions of exsolved volatiles. Due to the higher initial mass fractions of the MVP, these chambers do not necessarily reach undersaturation under the same conditions as deeper chambers but might require less volatile-rich initial chamber conditions or stronger recharge at the same initial magma conditions, see figure below.

Methods line 417: Please give the range of temperatures investigated.

Response:

We added a range of initial temperatures for the 15 vol% crystals case.

Method line 419: The listed CO₂ contents don't match the table.

Response:

Thank you for pointing this out, we corrected the typo in the text!

Method line 423: You don't give the composition for the recharge melt. Is it supposed to be a basalt? You give a melt density for the rhyodacite which is the most evolved end-member.

Response:

For model simplicity we actually assume the recharge magma to be of the same composition as the resident magma, therefore, we use the same density here. We simulate less evolved conditions of the recharging melt through higher temperatures and variable recharge contents of the inflowing magma.

Supplementary information 1 line 41: 0.09 is too high for your simulations, and also not what you present in Fig. 2C. Maybe 0.009? Also, not sure why these calculations were done on the basis of 10% initial crystallinity if the simulations are for 15%.

Response:

With these calculations we want to determine which of the processes in our 3 presented hypotheses has the largest effect on volatile resorption. We are not aiming to reproduce model outputs, as these would be subject to a large set of thermo-mechanical processes influencing the dynamics of the system, therefore isolating the individual effects would not be as easily doable. We use exemplary numbers here that are independent of our models. 0.09 was indeed a typo – we meant a $\Delta\varepsilon_x$ of 0.01, representing a change in crystallinity of 10%, thank you for spotting this!

Sincerely,
Janine Birnbaum

Reviewer #3 (Remarks on code availability):

I did not have access to the code repository.

Response:

Sorry for that! We added a link to our zenodo repository, but it might have gotten lost on the way. We will make sure to provide the access to the repository during re-submission.

REVIEWER COMMENTS

Reviewer #1 (Remarks to the Author):

I have read the revised version of that paper and found that the authors have correctly addressed the comments raised during the first round of review.

I would simply recommend the authors to change widespread process by common process in their abstract (I would even write ..likely a common process). I don't think we have the data to say that this paper would represent a paradigm shift in our perception of the mechanical evolution of magmatic reservoirs.

The excess sulfur conundrum documented for several explosive silicic eruptions is generally explained by the existence of a separate fluid phase, which, if true, shows that in those cases at least, volatile resorption did not play a crucial mechanical role.

I understand that to get published in journals of wide audience one has to show that the findings provided are of wide/global relevance but here it is mostly the attention to a under-appreciated mechanism which makes the paper appealing and interesting. In my opinion it is now the job of future studies to test that possibility before we can claim it is a general one.

Other than that I have no more comments to do and therefore recommend acceptance of this paper which addresses a mechanism which was so far not considered in the modelling reservoir evolution.

Bruno Scaillet

Response:

We thank reviewer 1 for their final comments. As suggested by the reviewer, we changed “widespread process” to “common process” in the abstract. We also agree that further studies will be required to test how widespread the resorption process really is. We also agree that resorption will likely not take place in all systems and also will not always reach undersaturation, as high recharge rates will be required, but we do see the appearance of resorption quite frequently in our models, which makes us curious to see future studies on this topic.

Reviewer #2 (Remarks to the Author):

I thank the authors for their engagement with my comments and questions. I appreciate the detail of their responses in the rebuttal and I am satisfied that they have made sufficient changes to the manuscript text and figures to address them. I am now happy to recommend this manuscript for publication, and hope that it sparks fruitful further discussions and new avenues of research by its readers.

Response:

We thank reviewer 2 for their time invested in our manuscript and are glad to hear that we could address their concerns appropriately.

Reviewer #3 (Remarks to the Author):

NCOMMS-25-51869A

Volatile resorption expedites eruption onset in large silicic systems

Keller, F., Townsend, M., Troch, J., Huber, C.

Main comments

Resorption drivers calculations:

1) Has this calculation now been removed from the methods?

Response:

We are not sure which calculation the reviewer is referring to here. As visible in the track changes recorded in the manuscript, we did not remove any calculations from the methods section. The equation for water mass conservation that elaborates on the drivers of resorption has always been in supplementary methods 1, where it is still available and we expanded on our approach.

2) The discussion of these is still a bit confused – going back and forth in order between the dilution of magma mixing and crystal melting

Response:

We are not sure what exactly the reviewer is referring to here. We worked quite significantly on improving the discussion part as this was also mentioned by other reviewers, who acknowledged that the manuscript has improved. In our opinion the discussion is clearer now, going from the most important processes hierarchically to the least important processes influencing resorption.

3) I think the new discussion is a cleaner in terms of the simplicity of the calculations, which is not per se a problem. But, I think there are two overlooked (competing) effects that need to be directly addressed in this discussion: you mix the water contents in process 2, but not the CO₂ contents which should have the opposite effect. It's fine to keep them very simple, treat them separately and show the effect of keeping water constant and essentially just adding some CO₂ (CO₂ flushing?) It seems from your supplementary data that the more CO₂ you add, the less resorption you get. Additionally, you do just the pressure effect, please also do just the temperature effect.

Response:

In the numerical model, we do account for both CO₂ and H₂O, it is only in the simplified scaling analysis that we focus on H₂O alone. We do this for two reasons: First, our model shows that resorption becomes less effective at higher CO₂ contents, largely because the total gas mass fraction increases, meaning more gas must be resorbed before the system reaches undersaturation. Second, we do not expect CO₂ to fundamentally control resorption dynamics in rhyolitic systems, which typically contain very little CO₂. For this reason, we focus on water primarily and believe that adding terms for CO₂ goes beyond the scope of this paper.

We further understand that the reviewer would like to see additional equations for temperature in the section explaining drivers for volatile resorption. As solubility is more pressure dependent than temperature dependent (see Liu et al., 2005: e.g. a 100 degree decrease in temperature for a water dominated gas phase causes a solubility increase of 0.4wt%), we believe that the main effect of increasing temperatures is melting crystals in the reservoir, which drives changes in the phase proportions as captured in process 3. Therefore, we would like to refrain from adding additional complexity to our equations, which might increase the difficulty for the readership to follow our thought process.

4) I asked on the first draft and I'm still confused. Why in your analysis do you only account for adding 1% of chamber volume through recharge when your undersaturated simulations add 10%? Eruptions occur for 10-2.4 km³/yr * 5x10³ yr = 20 km³ which is 4% of the total chamber volume, and you explore up to 10-2 km³/yr * 5x10³ yr = 50 km³ (10% of 500 km³). This will bring the effect closer to what you

get for 5% crystal melting (albeit that melt is completely anhydrous rather than 1 wt% water). Or is it that they erupt before getting there?

Response:

Indeed, in none of our simulations is a volume change of 10% achieved, as eruptions occur before such a volume change takes place. In our previous response we highlighted the new supplementary figure 3, which shows the volume change in one of the longest runs. In this run the max. volume change before eruption is 5 km³, which reflects 1% of the total volume. We would like to point out that the volume is not only controlled by mass influx but also the decrease in volume through resorption etc. Given these observations, we believe that modelling a max. volume change of 1% is reliable here and represented in our data.

Figure 3:

From this plotting it is not apparent to the reader what difference is from the resorption/exsolution and what is from the recharge rate. Can you do this in normalized time (time*recharge rate) which would just be added volume so that we can distinguish?

Response:

We don't think that this plot would actually change the current representation much. Runs 1 through 3 have the same recharge rate, therefore their relation will remain the same as presented in figure 3. The inclination of run 4 compared to the previous three might indeed change, but we don't think this will be of significant value for the reader. See the attached figure supporting our comment (due to the lower recharge rate, Run 4 is markedly shorter than runs 1-3).

In addition, we believe that changing the axis as suggested by the reviewer might actually make the figure harder to understand for non-modeler readership (see figure below with extended values on x-axis that are less intuitive than time in years). As Nature Communication is targeting a broad audience, we believe that keeping time instead of an equation on the x axis might be also preferable to address a broader readership.

Additionally, something looks odd to me in the calculation of average compressibility. Why is the average compressibility for run 3 between the value for runs 1 and 2 when it has uniformly higher vesicularity and lower pressure?

Response:

This is a good point. We double checked our calculations for the effective compressibility and realized that we calculated compressibility including during the time of eruption for this dataset, which can lead to wrong calculations. By calculating compressibility until the timestep right before eruption, we received values with a larger spread, even though the maximum change in compressibility is still low. We corrected the values in the figure.

Run 3 can be slightly less compressible than Run 2 even though it contains more water than Run 2. This is due to the dependence of solubility to thermodynamic and compositional parameters such as pressure and temperature. The temperature of our runs is calculated based on the initial volatile and crystal content of the respective runs and can therefore vary between individual runs. This is also the case here, Run 3 (867 °C) is slightly higher in temperature than Run 2 (856 °C), which can explain small differences in compressibility of 2×10^{-11} Pa⁻¹ as seen here.

I'm skeptical of this extrapolated curve. How was this fit? Run 2 also shows a certain acceleration despite remaining volatile saturated. If you adopt the scaled axis you could compare two runs that go from just volatile saturated to just undersaturated for a more direct comparison.

Response:

With these figures we primarily want to highlight the pressurization efficiency of runs with different recharge rates and water contents. For example, a run that is undergoing low recharge is exsolving, while pressurizing slower than all resorbing cases. Resorption and associated elevated pressurization on the other hand is connected to different water contents, where we see that lower water contents allow for resorption to undersaturation (which is not achieved in wetter run due to higher gas volume fractions) and allows for increased pressurization. The extrapolation of curve in Run 1 is secondary but included to show that undersaturation, if achieved, can lead to earlier eruption onset.

To extrapolate this curve, we extended the linear part of the curve before undersaturation has been achieved. We agree with the reviewer that curve 2 also shows inclination, however, as we are aiming to give examples of endmembers here that are secondary to the main message of the figure, and at the same time try to not overstate the early eruption onset (as also later pointed out by reviewer 3), we would like to keep the current extrapolation.

Compressibility calculation:

Since it is a critical parameter, how is the compressibility being calculated? It seems to me that from a volume-averaged-type formulation, the increase in compressibility at low volume fraction of exsolved volatiles into undersaturation should have a smooth, rather than stepped, increase in compressibility. Why would this produce a sharp change in the pressurization rate?

Also, the difference in compressibility is so small between the exsolving and volatile undersaturated runs (4×10^{-11} 1/Pa) which for 20 MPa of pressure should only be a volume difference of 0.08%. It seems to me that it's really just the volume reduction of the gas phase itself that's buffering the pressure, more than the "compressibility". I understand that in a bulk sense these are indistinguishable, but the discussion should be clear that this isn't about the instantaneous suspension compressibility if that's the case.

Response:

As mentioned in our previous responses, we did not calculate compressibility using a volume averaged approach but using the calculations from Townsend and Huber 2020 (see their supplement - <https://doi.org/10.1130/G47045.1>) that expand the derivative of mass density with respect to pressure. We reference this in both the figure captions of Figure 3 and 4.

The advantage of this calculation is that it considers the pressure dependence of volatile solubility and pressure-driven phase changes, which can capture changes in gas fraction better than a volume-average-

type formulation. As the appearance/ disappearance of a gas phase has significant impacts on the overall system compressibility we do see a sudden jump in compressibility and with that in pressurization.

Scaling argument (Fig. 4):

I still think it would be better to have a non-dimensional factor given that the pressure term should be combined with the rest. But interesting that including the volume change didn't help. That suggests to me that you haven't actually identified the driving process since compressibility alone can't explain your result!

Response:

We would like to re-emphasize that this is a commonly used scaling law and that adding a non-dimensional factor would probably not close the gap between the undersaturated and saturated cases. Therefore, we would like to keep the current version of the scaling law.

While we agree with the reviewer that additional processes besides compressibility (as calculated here) may be playing a role in changing the pressurization, we also believe that compressibility does play a significant role here as is shown in figure 4. We also acknowledge that the model includes many coupled and time-varying processes that will never be fully captured by a single scaling law. We have highlighted that a return to undersaturation is clearly important in driving up the pressurization rate compared to what we would expect in volatile saturated cases, and we invite readers to investigate the full effects of this process further.

See attached document for line-by-line comments.

Best regards,
Janine Birnbaum

Reviewer #3 (Remarks on code availability):

Given that the model is well-established, I didn't run the code, but I can see the files and inputs are available at the listed repository and all the data for the figures is provided in the supplementary material.

Line-by-line:

Lines 12-13: Noun disagreement between “volatile resorption” and “systems undergoing exsolution”.

Response:

Done.

Line 61: I would avoid the word “couple” here from a mathematics perspective and instead “we perform thermos-mechanical modelling tuned to/with inputs for geochemical data from Keller et al.”

Response:

We changed this to “we perform thermo-mechanical modelling informed by geochemical data...”.

Line 63: Omit “yet”.

Response:

Done.

Line 78: Are there melt inclusion volatile contents for this system that could hint at the volatile contents for the recharge? Is this a reasonable H₂O/CO₂ ratio?

Response:

From previous geochemical data, we know that recharge is of intermediate composition at Aso. As no MI data exists for the Aso-4 event itself, we tuned the volatile contents to the best of our knowledge to intermediate compositions, which are usually hotter and drier than rhyodacites/ rhyolites. In addition, we

tested over large, yet realistic ranges of these parameters to capture potential correlations. We find these are secondary to the recharge rate, as pointed out in the manuscript.

Line 86: This threshold for H₂O solubility needs to be paired with a CO₂ content for the mixed solubility model.

Response:

See table 1: initial concentrations for CO₂ used are 100 and 500 ppm CO₂. We do not find significant variations in solubility dependent on this small range of CO₂ starting compositions, crystallinity plays the major role as explained in the manuscript.

Line 88: recharge-induced variations in *pressure, temperature, and volatile content*; describe the direct impact on state variables and then the modeled effect on system characterization. You have not established through what mechanisms the resorption occurs.

Response:

In this section we are introducing our regime diagram, which captures the described variations using changes in crystal and gas fractions. We realized a typo here, we meant gas instead of melt volume fractions. In the next section of the manuscript, we explain in detail how pressure, temperature and volatile contents impact resorption mechanisms.

Line 90: It would be nice to directly state the regimes in the main text, not only in the figure.

Response:

Done.

Line 92: add “second boiling”.

Response:

Done.

Line 93: how would you describe this second regime? Just pressurization?

Response:

Yes, we believe this is a small fraction of cases where the recharge rates are high enough to pressurize the system without adding sufficient recharge to already heat the system. Such processes might be easier to achieve in smaller chambers.

Line 95-96: This question from the first set of reviews is unanswered. Is this regime physical/possible?

Response:

We would like to refer the reviewer to our answer in the previous reviews, we answered in detail there that we believe it could be possible under very specific conditions, which might be hard to achieve in nature but not impossible.

Line 97-100: This sentence doesn't tell me which recharge conditions can promote resorption – is it high temperatures, low H₂O, low CO₂?

Response:

We added these here.

Line 107: You've just told us that starting crystallinity has an important effect. What is the crystallinity of the Aso-4 system?

Response:

We would like to refer the reviewer to table 1, we tested both crystallinities of 2 and 15 vol% for Aso to cover the range of crystallinities observed.

Line 114: “variation in phase proportions of crystals, melt and MVP through influx of hotter recharge magma” – just crystal melting.

Response:

Done.

Line 120: Why calculate up to 25 MPa if you limit overpressure to 20 MPa?

Response:

This is a fair point. Mainly to have symmetry in the plots and have 5 points for each panel.

Line 121: “fixed” rather than “at zero”.

Response:

Done.

Line 127: Why discuss 3 before 2?

Response:

We explain the processes in hierarchical order of their contribution to resorption from the most important to the least important processes. Therefore, the order is process 1 – process 3 – process 2. We believe this is more intuitive to the reader and would like to keep it.

Line 134: Really the change in the melt volume.

Response:

*We are not sure what the reviewer is referring to here. If the reviewer is referring to “**The addition of anhydrous crystal melts and recharge melts dilutes the host magma’s..**”, we agree that the underlying mechanism is a change in the melt volume, however, we explain this in the previous sentence.*

Line 149: how can you assess the impact of pressurization in magma chamber pressurization? Please rephrase. Also melting rather than decreasing crystal fraction.

Response:

We believe we have addressed this point both in the manuscript and in our earlier responses related to Figure 3. The individual contributions are explicitly described and clarified in the main manuscript text, and we believe the current presentation is the most accurate and coherent way to convey our results. We would therefore prefer to keep the structure and content as it is.

Line 243: Isn’t the Liu et al. 2005 model for rhyolites? Can you speak to the difference from rhyodacite?

Response:

Indeed, Liu et al is calibrated for rhyolite, yet the difference between rhyolite and rhyodacite is negligible (rhyodacite being slightly lower in SiO₂ and alkalis than rhyolite) for such saturation models. In addition, to the best of our knowledge, no well-calibrated and well-established coupled CO₂-H₂O saturation model is established for specifically rhyodacite compositions, making a rhyolite calibrated model our best fit.

Line 267: Your crystallinity curves extend only to 1200 °C, but you have recharge at 1300 °C. Is this magma aphyric?

Response:

Yes, under these circumstances the 1300C magma would be considered aphyric, which is a realistic approximation given that the crystals crystallizing at these high temperatures are very few, close to 0.

Line 323: Is this the suspension density or the melt density?

Response:

This refers to the melt density. We clarified this in the manuscript.

Line 325: There is again disagreement between the listed CO₂ contents in the text here, what is shown in the figures, and what is presented in Table 1.

Response:

We caught a typo here stating 550 ppm instead of 500 ppm, we corrected this. We only present the lower recharge CO₂ and the highest recharge CO₂ here for easier visibility, the full range of CO₂ are given in supplementary fig. 1.

Line 373: For reference with very long author lists, please still give the first four authors followed by et al.

Response:

We changed this.

Line 512: Are the conditions for B different?

Response:

We assume the reviewer is referring to a typo in this line, which we corrected.

Figure 2: Here as well why “increasing melt fraction” rather than “melting”. And see main comments above, it would be nice to include in this figure the competing effects of temperature and CO₂ mixing. Panel B should be specified here that this is the effect of water dilution.

Response:

As pointed out in the main text of the manuscript the melt fraction is not only increasing due to crystal melting but also due to the addition of recharge magmas, which are lower in crystal fraction than the host magma. The addition of less crystal-rich magmas will also drive the crystal fraction of the host magma down. Therefore, it is a combination of crystal melting and crystal-poor recharge melt addition. We refer to the main comments regarding an answer to the second part of this comment.

Supplementary Figure 1: There’s a typo in the panel c legend “conent” instead of “content”. Also you have the CO₂ in panel D all in wt% when you use ppm elsewhere in the text.

Response:

Thank you for spotting the typo, we corrected this. The CO₂ in the panel D legend was in wt% due to space constraints but we updated this.

Supplementary methods 1:

Line 10: Define here again all the variables.

Response:

A definition of all variables is provided in the table at the end of this supplementary methods file.

Line 27: The notation has switched in the middle of the equation for the last two terms from $m_{eq}(P_0)$ to $m_{eq,0}$ and from $\rho_g(P_0)$ to $\rho_{g,0}$.

Response:

Thank you for spotting this typo, we corrected this.

Line 30: This equation uses the second notation from line 27, which is in conflict with the notation on lines 19-26.

Response:

We corrected this.

Line 30: The way this equation is written is a little chaotic: $\rho_m \partial m_{eq} / \partial P + \rho_m \partial m_{eq} / \partial P (-\epsilon_{chi0} - \epsilon_{g0})$ should be combined to $\rho_m \partial m_{eq} / \partial P (1 - \epsilon_{chi0} - \epsilon_{g0})$ to preserve the structure of the original equation and combine terms. You can also preserve more of the original structure by multiplying by keeping some signs more consistent and

factoring:

$$\Delta \varepsilon_g = \left(\rho_m \frac{\partial m_{eq}}{\partial P} (1 - \varepsilon_{\chi, 0} - \varepsilon_{g, 0}) + \frac{\partial \rho_g}{\partial P} \varepsilon_{g, 0} \right) \Delta P - \Delta \varepsilon_{\chi} \rho_m \left(m_{eq, 0} + \frac{\partial m_{eq}}{\partial P} \Delta P \right) - \left(\frac{\partial M_w}{\partial V_0} - \frac{\partial M_w}{\partial V_i} \right) \rho_m m_{eq, 0} - \rho_{g, 0} + \left(\rho_m \frac{\partial m_{eq}}{\partial P} - \frac{\partial \rho_g}{\partial P} \right) \Delta P$$

Response:

We updated the equation structure as suggested by the reviewer.

Line 43: In this formulation the recharge density must be the same as the chamber melt density, which is 2250 kg/m³ not 2400 kg/m³. You've restricted the compositions to be the same, the densities should be as well.

Response:

For our code this is correct. Yet the examples we are bringing on here are unrelated to the code, so using a higher recharge density is more realistic here and we would like to keep it that way.

Previous round of review:

There was a question from reviewer 1 about the host rock compressibility, which I was also curious about. The value was given in the response, but not added to the text as far as I could tell. Fine to go in the methods or supplement, but it should be there. Better if it can be directly compared to the magma compressibility.

Response:

We believe adding this value to the methods section is not necessary here. The value for the bulk modulus is available in the code provided in the supplements and does not play a crucial role in the model setup.

Hornblende in equilibrium with the melt: I'm not a mineralogist, but I would be surprised to find that hornblende was in equilibrium with a *water undersaturated melt*.

Response:

The main question arising from the papers cited by reviewer 1 is whether the hornblende grew in the Aso-4 magmas or if it was entrained from an underlying mush. This question is mainly geared to how reliable the ages provided by K/Ar dating are, it seems like the reviewer did not aim to comment on this issue. Resorption is a process on much shorter timescales and we still believe that the hornblende grew from the Aso-4 magmas due to the reasons stated in the previous reviews before the magma became undersaturated shortly before eruption.

In addition, we want to point out that indeed hornblende grows in hydrous magmas, which does not immediately mean that the magma was water saturated. There is no evidence that we are aware of that hornblende only grows in water-saturated environments.

I agree with reviewer 2 that the early eruption onset is not the main (or most robust) claim of the paper and an alternative title focusing on that recharge might favor resorption is stronger. I undersaturation really the critical point either?

Response:

We would like to bring our answers to reviewer 2 to the attention of the reviewer. There we explain in detail that we agree with reviewer 2 that undersaturation is not the critical point, and why we would like to keep the current title.

Response to Referee's

REVIEWERS' COMMENTS

Reviewer #1 (Remarks to the Author):

I have had now time to read carefully the last version of the paper, as well as the accompanying rebuttal letter.

It is my opinion that the authors have done a good job in clearly and carefully addressing/considering the majority of the comments raised during the several rounds of reviews.

There are obviously still open questions but in my opinion they pertain to the scientific debate. Those questions may even promote future work on the subject, in particular additional modelling efforts, which will deal with more complete/complex treatment of the physical process addressed here (for instance what would be the role of the geometry of reservoir, or its depth, which may affect the tensile strength of host rocks).

I therefore recommend acceptance.

Bruno Scaillet

Our reply:

We would like to sincerely thank Reviewer 1 for their continued engagement with our manuscript. We truly appreciate the time they took to review this latest version. We are glad that our efforts to address the previous concerns were helpful and that the reviewer found the revised manuscript to be improved.

We absolutely agree that there are still some open questions about the process of volatile resorption especially due to the complexity of magmatic systems and hope that this might inspire future research. To highlight this more carefully in the manuscript, we added a couple of sentences addressing the model uncertainty at the end of the manuscript in ll. 244.

Thank you for your time again!

Franziska Keller and co-authors.

NCOMMS-25-51869A

Volatile resorption expedites eruption onset in large silicic systems

Keller, F., Townsend, M., Troch, J., Huber, C.

Main comments

Resorption drivers calculations:

- 1) Has this calculation now been removed from the methods?
- 2) The discussion of these is still a bit confused – going back and forth in order between the dilution of magma mixing and crystal melting
- 3) I think the new discussion is a cleaner in terms of the simplicity of the calculations, which is not per se a problem. But, I think there are two overlooked (competing) effects that need to be directly addressed in this discussion: you mix the water contents in process 2, but not the CO₂ contents which should have the opposite effect. It's fine to keep them very simple, treat them separately and show the effect of keeping water constant and essentially just adding some CO₂ (CO₂ flushing?) It seems from your supplementary data that the more CO₂ you add, the less resorption you get. Additionally, you do just the pressure effect, please also do just the temperature effect.
- 4) I asked on the first draft and I'm still confused. Why in your analysis do you only account for adding 1% of chamber volume through recharge when your undersaturated simulations add 10%? Eruptions occur for $10^{-2.4} \text{ km}^3/\text{yr} * 5 \times 10^3 \text{ yr} = 20 \text{ km}^3$ which is 4% of the total chamber volume, and you explore up to $10^{-2} \text{ km}^3/\text{yr} * 5 \times 10^3 \text{ yr} = 50 \text{ km}^3$ (10% of 500 km³). This will bring the effect closer to what you get for 5% crystal melting (albeit that melt is completely anhydrous rather than 1 wt% water). Or is it that they erupt before getting there?

Figure 3:

From this plotting it is not apparent to the reader what difference is from the resorption/exsolution and what is from the recharge rate. Can you do this in normalized time (time*recharge rate) which would just be added volume so that we can distinguish?

Additionally, something looks odd to me in the calculation of average compressibility. Why is the average compressibility for run 3 between the value for runs 1 and 2 when it has uniformly higher vesicularity and lower pressure?

I'm skeptical of this extrapolated curve. How was this fit? Run 2 also shows a certain acceleration despite remaining volatile saturated. If you adopt the scaled axis you could compare two runs that go from just volatile saturated to just undersaturated for a more direct comparison.

Compressibility calculation:

Since it is a critical parameter, how is the compressibility being calculated? It seems to me that from a volume-averaged-type formulation, the increase in compressibility at low volume fraction of exsolved volatiles into undersaturation should have a smooth, rather than stepped, increase in compressibility. Why would this produce a sharp change in the pressurization rate?

Also, the difference in compressibility is so small between the exsolving and volatile undersaturated runs ($4e-11$ 1/Pa) which for 20 MPa of pressure should only be a volume difference of 0.08%. It seems to me that it's really just the volume reduction of the gas phase itself that's buffering the pressure, more than the "compressibility". I understand that in a bulk sense these are indistinguishable, but the discussion should be clear that this isn't about the instantaneous suspension compressibility if that's the case.

Scaling argument (Fig. 4):

I still think it would be better to have a non-dimensional factor given that the pressure term should be combined with the rest. But interesting that including the volume change didn't help. That suggests to me that you haven't actually identified the driving process since compressibility alone can't explain your result!

Line-by-line:

Lines 12-13: Noun disagreement between "volatile resorption" and "systems undergoing exsolution".

Line 61: I would avoid the word "couple" here from a mathematics perspective and instead "we perform thermos-mechanical modelling tuned to/with inputs for geochemical data from Keller et al."

Line 63: Omit "yet".

Line 78: Are there melt inclusion volatile contents for this system that could hint at the volatile contents for the recharge? Is this a reasonable H₂O/CO₂ ratio?

Line 86: This threshold for H₂O solubility needs to be paired with a CO₂ content for the mixed solubility model.

Line 88: recharge-induced variations in *pressure, temperature, and volatile content*; describe the direct impact on state variables and then the modeled effect on system characterization. You have not established through what mechanisms the resorption occurs.

Line 90: It would be nice to directly state the regimes in the main text, not only in the figure.

Line 92: add "second boiling".

Line 93: how would you describe this second regime? Just pressurization?

Line 95-96: This question from the first set of reviews is unanswered. Is this regime physical/possible?

Line 97-100: This sentence doesn't tell me which recharge conditions can promote resorption – is it high temperatures, low H₂O, low CO₂?

Line 107: You've just told us that starting crystallinity has an important effect. What is the crystallinity of the Aso-4 system?

Line 114: "variation in phase proportions of crystals, melt and MVP through influx of hotter recharge magma" – just crystal melting.

Line 120: Why calculate up to 25 MPa if you limit overpressure to 20 MPa?

Line 121: “fixed” rather than “at zero”.

Line 127: Why discuss 3 before 2?

Line 134: Really the change in the melt volume.

Line 149: how can you assess the impact of pressurization in magma chamber pressurization? Please rephrase. Also melting rather than decreasing crystal fraction.

Line 243: Isn't the Liu et al. 2005 model for rhyolites? Can you speak to the difference from rhyodacite?

Line 267: Your crystallinity curves extend only to 1200 °C, but you have recharge at 1300 °C. Is this magma aphyric?

Line 323: Is this the suspension density or the melt density?

Line 325: There is again disagreement between the listed CO₂ contents in the text here, what is shown in the figures, and what is presented in Table 1.

Line 373: For reference with very long author lists, please still give the first four authors followed by et al.

Line 512: Are the conditions for B different?

Figure 2: Here as well why “increasing melt fraction” rather than “melting”. And see main comments above, it would be nice to include in this figure the competing effects of temperature and CO₂ mixing. Panel B should be specified here that this is the effect of water dilution.

Supplementary Figure 1: There's a typo in the panel c legend “conent” instead of “content”. Also you have the CO₂ in panel D all in wt% when you use ppm elsewhere in the text.

Supplementary methods 1:

Line 10: Define here again all the variables.

Line 27: The notation has switched in the middle of the equation for the last two terms from $m_{eq}(P_0)$ to $m_{eq,0}$ and from $\rho_g(P_0)$ to $\rho_{g,0}$.

Line 30: This equation uses the second notation from line 27, which is in conflict with the notation on lines 19-26.

Line 30: The way this equation is written is a little chaotic: $\rho_m \partial m_{eq} / \partial P + \rho_m \partial m_{eq} / \partial P (-\epsilon_{\chi,0} - \epsilon_{g,0})$ should be combined to $\rho_m \partial m_{eq} / \partial P (1 - \epsilon_{\chi,0} - \epsilon_{g,0})$ to preserve the structure of the original equation and combine terms. You can also preserve more of the original structure by multiplying by keeping some signs more consistent and factoring:

$$\Delta \epsilon_g = \frac{\left(\rho_m \frac{\partial m_{eq}}{\partial P} (1 - \epsilon_{\chi,0} - \epsilon_{g,0}) + \frac{\partial \rho_g}{\partial P} \epsilon_{g,0} \right) \Delta P - \Delta \epsilon_{\chi} \rho_m \left(m_{eq,0} + \frac{\partial m_{eq}}{\partial P} \Delta P \right) - \left(\frac{\partial M_{w,0}}{\partial V_0} - \frac{\partial M_{w,i}}{\partial V_i} \right)}{\rho_m m_{eq,0} - \rho_{g,0} + \left(\rho_m \frac{\partial m_{eq}}{\partial P} - \frac{\partial \rho_g}{\partial P} \right) \Delta P}$$

Line 43: In this formulation the recharge density must be the same as the chamber melt density, which is 2250 kg/m³ not 2400 kg/m³. You've restricted the compositions to be the same, the densities should be as well.

Previous round of review:

There was a question from reviewer 1 about the host rock compressibility, which I was also curious about. The value was given in the response, but not added to the text as far as I could tell. Fine to go in the methods or supplement, but it should be there. Better if it can be directly compared to the magma compressibility.

Hornblende in equilibrium with the melt: I'm not a mineralogist, but I would be surprised to find that hornblende was in equilibrium with a *water undersaturated melt*.

I agree with reviewer 2 that the early eruption onset is not the main (or most robust) claim of the paper and an alternative title focusing on that recharge might favor resorption is stronger. I undersaturation really the critical point either?